# Modulation of Fibroblast Activity via Vitamin D_3_ Is Dependent on Tumor Type—Studies on Mouse Mammary Gland Cancer

**DOI:** 10.3390/cancers14194585

**Published:** 2022-09-21

**Authors:** Natalia Łabędź, Martyna Stachowicz-Suhs, Mateusz Psurski, Artur Anisiewicz, Joanna Banach, Aleksandra Piotrowska, Piotr Dzięgiel, Adam Maciejczyk, Rafał Matkowski, Joanna Wietrzyk

**Affiliations:** 1Department of Experimental Oncology, Hirszfeld Institute of Immunology and Experimental Therapy, Weigla 12, 53-114 Wroclaw, Poland; 2Division of Histology and Embryology, Department of Human Morphology and Embryology, Faculty of Medicine, Wroclaw Medical University, Chałubińskiego 6a, 50-368 Wroclaw, Poland; 3Department of Oncology, Wroclaw Medical University, Pl. Ludwika Hirszfelda 12, 53-413 Wroclaw, Poland; 4Lower Silesian Oncology, Pulmonology and Hematology Center, Pl. Ludwika Hirszfelda 12, 53-413 Wroclaw, Poland

**Keywords:** vitamin D, calcitriol, breast cancer, cancer-associated fibroblasts, CAFs, angiogenesis

## Abstract

**Simple Summary:**

This study, which was conducted in healthy mice and mice bearing three mouse mammary gland cancers—4T1, 67NR, and E0771—showed that the divergent effects of vitamin D_3_ supplementation (5000 IU) or deficiency (100 IU of vitamin D_3_) observed in healthy mice led to the formation of various body microenvironments depending on the mouse strain. Developing tumors themselves modified the microenvironments by producing higher concentrations of osteopontin, SDF-1 (4T1), TGF-β (4T1 and E0771), CCL2, VEGF, FGF23 (E0771), and IL-6 (67NR), which influences the response to vitamin D_3_ supplementation/deficiency and calcitriol administration and leads to enhanced/decreased activation of lung fibroblasts and modulation of tumor tissue blood flow.

**Abstract:**

Vitamin D_3_ and its analogs are known to modulate the activity of fibroblasts under various disease conditions. However, their impact on cancer-associated fibroblasts (CAFs) is yet to be fully investigated. The aim of this study was to characterize CAFs and normal fibroblasts (NFs) from the lung of mice bearing 4T1, 67NR, and E0771 cancers and healthy mice fed vitamin-D_3_-normal (1000 IU), -deficient (100 IU), and -supplemented (5000 IU) diets. The groups receiving control (1000 IU) and deficient diets (100 IU) were gavaged with calcitriol (+cal). In the 4T1-bearing mice from the 100 IU+cal group, increased NFs activation (increased α-smooth muscle actin, podoplanin, and tenascin C (TNC)) with a decreased blood flow in the tumor was observed, whereas the opposite effect was observed in the 5000 IU and 100 IU groups. CAFs from the 5000 IU group of E0771-bearing mice were activated with increased expression of podoplanin, platelet-derived growth factor receptor β, and TNC. In the 100 IU+cal group of E0771-bearing mice, a decreased blood flow was recorded with decreased expression of fibroblast growth factor 23 (FGF23) and C-C motif chemokine ligand 2 (CCL2) in tumors and increased expression of TNC on CAFs. In the 67NR model, the impact of vitamin D_3_ on blood flow or CAFs and lung NFs was not observed despite changes in plasma and/or tumor tissue concentrations of osteopontin (OPN), CCL2, transforming growth factor-β, vascular endothelial growth factor, and FGF23. In healthy mice, divergent effects of vitamin D_3_ supplementation/deficiency were observed, which lead to the creation of various body microenvironments depending on the mouse strain. Tumors developing in such microenvironments themselves modified the microenvironments by producing, for example, higher concentrations of OPN and stromal-cell-derived factor 1 (4T1), which influences the response to vitamin D_3_ supplementation/deficiency and calcitriol administration.

## 1. Introduction

The tumor microenvironment, which includes fibroblasts, vascular endothelial cells, and immune cells, is an important component of cancer and affects its progression, metastasis, and sensitivity to therapies. Vitamin D_3_—mainly its active form, calcitriol, 1,25-dihydroxycholecalciferol—can influence almost every cell in the body, including the tumor-building ones [1]. Thus, although cancer cells (e.g., murine cancer cell lines) are not sensitive to the antiproliferative activity of vitamin D_3_ in vitro (despite the presence of the vitamin D receptor, VDR), they show significant sensitivity to vitamin D derivatives in vivo. One such example is 4T1 mouse breast cancer cells. Previous studies have shown that calcitriol and its analogs may increase the rate of tumor metastasis [2,3] or accelerate the progression of primary tumors in young mice [4]. However, in aged, ovariectomized animals (as a postmenopausal model), these compounds inhibited 4T1 tumor metastasis [5]. Besides the age of mice, other factors may also affect the outcome of the in vivo treatment of mammary gland tumors with vitamin D compounds. The abovementioned observations were from experiments in mice in which the therapy was started when tumors were already formed. On the contrary, an antimetastatic effect was observed in young mice bearing 4T1 tumors when calcitriol treatment was started prior to tumor inoculation [6]. The beneficial effect of vitamin D analogs in young mice was also evident when these substances were used in combination therapy with cyclophosphamide [7]. In the case of E0771, another mouse mammary gland cancer model that is also not sensitive to the antiproliferative effect of calcitriol [3], the antitumor effect of cholecalciferol was observed in normal young mice, whereas protumor activity was observed in obese mice. This effect was related to the impact of the treatment on tumor-infiltrating CD8+ T lymphocytes which are increased in number in normal mice and decreased in obese mice [8].

In recent clinical studies much attention has been paid on the effects of vitamin D in the development of various cancers, including breast cancer. These studies focused mainly on the relationship between tumor progression and plasma 25(OH)D concentrations (the most studied metabolite of vitamin D), vitamin D supplementation or the expression of various proteins related to vitamin D activity or metabolism (recently reviewed in [9]). However, based on these studies it is difficult to draw firm conclusions concerning beneficial or adverse effects of vitamin D on breast cancer development. For example, a systematic review and meta-analysis by Hossain et al. showed an inverse relationship between vitamin D intake and breast cancer occurrence along with a direct relationship between vitamin D deficiency and breast cancer risk [10]. Other studies reported the occurrence of vitamin D deficiency in newly diagnosed breast cancer patients [11]. They also indicate that vitamin D supplementation significantly reduced total cancer mortality but did not affect total cancer incidence [12]. On the other hand, e.g., O’Connor et al. indicated the lack of the relationship between vitamin D and breast cancer [13]. Moreover, Kanstrup et al. [14] found an inverse correlation between 25(OH)D plasma concentrations and breast cancer survival, however, this study also indicated poorer breast cancer survival among patients with high 25(OH)D concentrations [14]. Likewise, Ganji et al. reported an association of high 25(OH)D plasma concentrations (75–100 nmol/L) with greater risk of breast cancer in postmenopausal women [14,15]. Therefore, controlled trials are required to verify the data concerning vitamin D benefits for breast cancer patients.

Cancer-associated fibroblasts (CAFs) resembling myofibroblasts—activated spindle-shaped fibroblasts—are the major cellular component in the tumor microenvironment [16]. The stimuli produced by tumor cells and stromal cells, such as fibroblasts, in the tumor microenvironment can activate fibroblasts and contain, among others, interleukin-1, interleukin-6 (IL-6), bone morphogenetic protein, sonic hedgehog, reactive oxygen species, transforming growth factor-β (TGF-β), platelet-derived growth factor, and tumor necrosis factor [17]. The activation of fibroblasts by these stimuli may result in quiescent, tumor-restraining, and tumor-promoting CAFs [18,19,20]. Orimo et al. showed that coimplantation of the tumor xenograft model with CAFs harvested from the breast carcinomas of patients promoted higher growth of breast cancer cells compared with coimplantation with normal mammary fibroblasts derived from the same patients [21]. CAFs promote tumor growth through the secretion of stromal-cell-derived factor 1 (SDF-1) and angiogenesis by recruiting endothelial progenitor cells into tumor tissue [21]. Thus, some studies have proposed targeting CAFs as a treatment option for triple-negative breast cancer [22]. Clinical retrospective studies on breast cancer patients focusing on the expression of caveolin-1, which acts as a tumor-suppressing molecule in CAFs, showed 72 months of cancer-specific survival in caveolin-1-positive patients, whereas only 29.5 months in caveolin-1-negative patients [23].

Osteopontin (OPN) is an important protein engaged in the crosstalk between cancer cells and stromal fibroblasts. OPN secreted by breast cancer cells induces differentiation of fibroblasts to myofibroblasts. Furthermore, OPN-driven CAFs secrete SDF-1, which in turn triggers epithelial-to-mesenchymal transition in tumor cells [24]. As reported in previous studies, calcitriol and other agonists of VDR stimulate the transcription of the OPN gene (*Spp1*) or secretion of OPN [25,26]. In our previous in vitro study, calcitriol and its analogs stimulated OPN secretion in 67NR mouse mammary gland cancer cells, which are sensitive to proliferation inhibition by calcitriol, but not in 4T1 cells [2]. It was also found that in vitro stimulation by calcitriol and its analogs led to increased OPN secretion by normal mouse fibroblast cells BALB/3T3, but not by the murine macrophage cell line RAW 264.7 [27]. Although much research has been conducted investigating the effects of vitamin D on fibroblasts in various types of diseases or physiological conditions [28,29,30,31], its effect on CAFs remains to be elucidated. Studies on fibroblasts derived from colon cancer [32,33], pancreatic cancer [34,35], and breast cancer [36,37] patients have indicated that calcitriol modulates gene expression and inhibits the protumoral properties of CAFs. Therefore, in the present study, we analyzed the effects of calcitriol treatment and vitamin D_3_ (cholecalciferol) deprivation or supplementation on tumor and lung fibroblasts in three mouse mammary gland cancer models (4T1, 67NR, and E0771) with different metastatic potential.

## 2. Materials and Methods

### 2.1. Cells and Tissues Harvested from Mice Bearing Mammary Gland Tumors

Fibroblasts were isolated from lung and tumor tissues of mice bearing 4T1, 67NR, and E0771 tumors and of healthy BALB/c and C57BL/6 mice. Data regarding tumor growth, metastasis, and vitamin D metabolism in these mice have been previously published [3]. Origin of the cell lines used: 4T1 cells-the American Type Culture Collection (ATCC, Rockville, MD, USA); 67NR (nonmetastatic counterparts of 4T1) Barbara Ann Karmanos Cancer Institute (Detroit, MI, USA); E0771 cell line [38], gifted by Dr. Andreas Möller (School of Medicine, University of Queensland; Tumour Microenvironment Laboratory, QIMR Berghofer Medical Research Institute, Herston, Queensland, Australia).

The complete in vivo experimental design has been described earlier [3]. The animal study protocol was approved by the first Local Committee for Experiments with the Use of Laboratory Animals, Hirszfeld Institute of Immunology and Experimental Therapy, Wroclaw, Poland (permission number: 66/2018, 18 July 2018). Mice were divided into groups and fed ad libitum with an AIN67 synthetic diet (ZooLab, Sedziszow, Poland) for 6 weeks. The diet fed to groups had a varied content of vitamin D_3_ as follows: control amount of vitamin D_3_ (1000 IU/kg), supplemented vitamin D_3_ (5000 IU/kg), and deficient vitamin D_3_ (100 IU/kg). After 6 weeks (day 0), some mice were implanted orthotopically with tumor cells (1 × 104, 2 × 105, and 5 × 104 viable 4T1, 67NR, and E0771 cells, respectively), and the same diet was continued. Seven days after the implantation of tumor cells, calcitriol administration (1 µg/kg per os by gavage; p.o.) was started, which continued thrice a week in the groups receiving food with a normal, control amount of vitamin D_3_ (1000 IU) and in the groups receiving a vitamin-D_3_-deficient diet (100 IU). On day 23 (C57BL/6) or 28 (BALB/c) after the inculcation of cancer cells, mice were subjected to isoflurane anesthesia and injected with buprenorphine (0.1 mg/kg of body weight) analgesia, subjected to blood collection, and then killed (Figure 1).

### 2.2. Blood Flow Assessment

Tumor blood perfusion analysis was performed on day 21 (4T1 and 67NR models) or on day 19 (E0771 model) using the MicroMarker™ contrast agent (VisualSonics, Toronto, ON, Canada) and Vevo 2100 ultrasound imaging system (VisualSonics). Mice were anesthetized by continuous administration of 2–3% (*v*/*v*) isoflurane (Baxter, Deerfield, Germany) in synthetic air (600 mL/min) and immobilized. The tumor area was enclosed with air-bubble-free gel, and the central cross-section of the tumor was visualized in the transverse plane using an MS250 scanhead (VisualSonics). Then, 100 μL of the contrast agent was injected intravenously and the first imaging sequence (bolus) was recorded (ca. 15 fps, at least 1000 frames) after the contrast signal in the tumor tissue reached the steady state (ca. 50 s). Next, using the burst mode, contrast microbubbles were destroyed, and the second imaging sequence (replenishment) was recorded. Mice were maintained in a warm environment until fully awakened. The data were analyzed using the Vevo LAB 1.7.1 Software with VevoCQ modality (VisualSonics).

### 2.3. Tissue Microarrays 

Tissue microarrays were prepared using TMA Grand Master (3DHistech, Budapest, Hungary), in accordance with the manufacturer’s instructions. The representative spots for tissue microarrays were selected by a pathologist. Three core punches of 1.5 mm diameter from each tumor block were transferred into the recipient paraffin block.

### 2.4. Immunohistochemical Staining of Tissues

Using the automated immunostainer device Autostainer Link 48 (Dako, Glostrup, Denmark), immunohistochemistry analysis was performed on 4 µm formalin-fixed paraffin-embedded tissue microarray sections. To deparaffinize, rehydrate, and unmask the epitopes, the slides were boiled in EnVision™ (Santa Clara, CA, USA) FLEX Target Retrieval Solution of a high pH for 20 min at 97 °C using PT-Link (both from Dako). The endogenous peroxidase activity was blocked using the EnVision FLEX Peroxidase-Blocking Reagent (Dako). Then, the primary antibody against collagen type I alpha 1 chain (COL1A1) (dilution 1:1600, cat.no. PA5-86949; Invitrogen, Waltham, MA, USA) or against the smooth muscle actin (ready-to-use, cat. no. IR611; Dako, Glostrup, Denmark) were applied for 20 min at room temperature. Afterward, the slides were incubated with EnVision FLEX/HRP (Dako) for (20 min). Subsequently, the sections were incubated for 10 min at room temperature with the substrate for peroxidase, diaminobenzidine. Additionally, using the EnVision FLEX Hematoxylin (Dako), all slides were counterstained for 5 min. After dehydration in graded ethanol concentrations (70%, 96%, absolute) and in xylene, all slides were covered with coverslips in the SUB-X Mounting Medium. Two pathologists evaluated the expression of studied proteins with a routinely used immunoreactive scale of Remmele and Stegner [39], which is presented in Table 1.

### 2.5. Isolation of Lung and Tumor Fibroblasts

Harvested tissues were transferred to a 6 cm tissue culture dish, mechanistically chopped into ~1 mm pieces, and digested for 1 h at 37 °C with gentle shaking. The digestion solution was composed of phosphate-buffered saline containing calcium and magnesium ions (PBS Ca^2+^ Mg^2+^; HIIET, Wroclaw, Poland), 1 mg/mL collagenase IV (collagenase from *Clostridium histolyticum*; Sigma-Aldrich, Saint-Louis, MO, USA), and 1 mg/mL DNase I (Roche, Basel, Switzerland). Digested lung homogenates were centrifuged for 7–10 min (4 °C, 350× *g*) and washed thrice with a normal fibroblast (NF) medium, Dulbecco’s modified Eagle’s medium (DMEM; Gibco, Grand Island, NY, USA), containing 10% (*v*/*v*) of fetal bovine serum (FBS), 1% (*v*/*v*) nonessential amino acids, 4.0 mM l-glutamine, 100 μg/mL streptomycin (all Sigma-Aldrich, Saint-Louis, MO, USA), and 100 U/mL penicillin (Polfa Tarchomin S.A., Warsaw, Poland). Cell pellets were resuspended in FBS (Sigma-Aldrich, Saint-Louis, MO, USA) with 10% (*v*/*v*) DMSO (Sigma-Aldrich, Saint-Louis, MO, USA) and frozen for further analysis. After the digestion process was completed, tumor homogenates were passed through a strainer with a mesh size of 70 μm (Greiner Bio-One, Kremsmünster, Austria) and then washed with fresh PBS containing 2% (*v*/*v*) FBS. Cell suspension was centrifuged for 7–10 min (4 °C, 350× *g*), followed by erythrocytes lysis. In brief, pellets were resuspended with 1 mL of lysis buffer (Sigma-Aldrich, Saint-Louis, MO, USA) and shaken for 1 min; then, PBS with serum was added and the suspension was centrifuged for 7–10 min (4 °C, 350× *g*). The cell pellets were resuspended in 5 mL of fresh PBS with FBS, and cell quantity was assessed by counting in a Bürker chamber in trypan blue solution (0.4% (*w*/*v*); Sigma-Aldrich, Saint-Louis, MO, USA).

### 2.6. Magnetic Separation

For magnetic separation, whole-cell suspensions (three mice from one group were pooled together for one separation) were used. CAFs were isolated using Anti-Fibroblast MicroBeads (Miltenyi Biotec, Auburn, CA, USA), according to the manufacturer’s protocol. In brief, centrifuged pellets (7–10 min, 4 °C, 350× *g*) were resuspended in the separation buffer containing PBS of pH 7.2, 0.5% (*v*/*v*) bovine serum albumin, 2 mM ethylene diamine tetra-acetic acid (Sigma-Aldrich, Saint-Louis, MO, USA), and TruStain FcX (antimouse CD16/CD_3_2) antibody (BioLegend, San Diego, CA, USA) and then incubated for 10 min at 4 °C in order to block the Fc receptors (0.1 µg/100 µL volume). After blocking, magnetic beads were added (20 µL per 10^6^ of total cells) to the cells and incubated for 30 min in the dark at room temperature. Subsequently, 1 mL of separation buffer was added to the cells and centrifuged (7–10 min, 4 °C, 350× *g*). The pellets were resuspended in 1 mL of separation buffer and applied onto activated MS columns (Miltenyi Biotec, Auburn, CA, USA) placed in the magnetic field of the MiniMACS Separator (Miltenyi Biotec, Auburn, CA, USA). After three washes with 500 µL of separation buffer, the columns were transferred into new sterile 10 mL tubes, and cells were flushed into collection tubes. The collected cells were counted using a Bürker chamber in trypan blue solution (0.4% (*w*/*v*)) and used for further analysis.

### 2.7. Flow Cytometry Analysis of Isolated Fibroblasts

CAFs isolated from tumor tissues were analyzed immediately after the separation. Lung fibroblasts from tumor-bearing mice were processed directly after thawing, and those from healthy mice were subjected to cytometry analysis after stimulation with TGF-β1 or cancer-cell-conditioned medium. To assess cell viability, cell suspensions of 3–5 × 10^4^ cells per sample were resuspended in pure PBS and incubated with eBioscience™ Fixable Viability Dye eFluor™ 780 (Invitrogen, Waltham, MA, USA) for 30 min at 4 °C. Cell surface markers were stained extracellularly in the FACS buffer (2% (*v*/*v*) FBS in FBS) for 30 min at 4 °C. To carry out intracellular staining, cells were fixed in a fixation buffer (BioLegend, San Diego, CA, USA) for 20 min at room temperature, subsequently washed, and permeabilized using the Intracellular Staining Perm Wash Buffer (BioLegend, San Diego, CA, USA) thrice (7 min, 20 °C, 350× *g*). The samples resuspended in the FACS buffer were read using a BD LSR Fortessa cytometer with FACSDiva V8.0.1 software (BD Biosciences, Franklin Lakes, NJ, USA). Cytometric analysis was performed for four replicates (pooled from three mice each). For each of the markers, the median fluorescence intensity (MFI) of stained cells in relation to the isotype control and the percentage of positive cells were determined. Following antibodies were used for CAF staining: α-smooth muscle actin (α-SMA)-PE (Abcam, Cambridge, UK), CD_3_1-BV-421, CD90.2-PerCP/Cy5.5, EpCam-FITC, platelet-derived growth factor receptor β (PDGFRβ)-APC, Podoplanin-PE-Cy7 (all Biolegend, San Diego, CA, USA), and Tenascin C (TNC)-Alexa Fluor 700 (Novus Biologicals, Centennial, CO, USA). For NF staining CD90.2 was replaced with CD45 with the same dye (BD Biosciences, San Jose, CA, USA). 

### 2.8. Preparation of Conditioned Media (CM) from 4T1, 67NR, and E0771 Cell Cultures

Cancer cell lines (4T1, 67NR, and E0771) were grown on appropriate media on 10 cm tissue culture dishes (4T1 in RPMI 1640 medium with GlutaMAX, supplemented with 10% (*v*/*v*) FBS HyClone (both Thermo Fisher Scientific, Waltham, MA, USA), 1 mM sodium pyruvate, 3.5 g/L glucose, 100 μg/mL streptomycin (all Sigma-Aldrich, Saint-Louis, MO, USA), and 100 U/mL penicillin (Polfa Tarchomin S.A., Warsaw, Poland); 67NR in DMEM (Gibco, Grand Island, NY, USA) supplemented with 10% (*v*/*v*) calf bovine serum (ATCC, Manassas, VA, USA), 1× nonessential amino acids, 2.0 mM l-glutamine, 100 μg/mL streptomycin (all Sigma-Aldrich, Saint-Louis, MO, USA), and 100 U/mL penicillin (Polfa Tarchomin S.A., Warsaw, Poland); E0771 in DMEM (Gibco, Grand Island, NY, USA) supplemented with 10% (*v*/*v*) FBS HyClone, 2.0 mM l-glutamine, 100 μg/mL streptomycin (all Sigma-Aldrich, Saint-Louis, MO, USA), and 100 U/mL penicillin (Polfa Tarchomin S.A., Warsaw, Poland)). The dishes were washed with PBS after the cells reached confluence, and cells were incubated for 24 h with an appropriate medium without serum. Then, CM were collected, centrifuged (7 min, 4 °C, 400× *g*), and applied in a 1:1 ratio (CM:NFs medium) for 72 h on NF cultures for stimulation.

### 2.9. Stimulation of Lung Fibroblasts from Healthy Mice with TGF-β and Mammary Gland Cancer CM

NFs from healthy mice were thawed in NF medium onto a 10 cm tissue culture dish and cultured at 37 °C in a humidified atmosphere with 5% (*v*/*v*) CO_2_. After reaching >60% confluence, cells were transferred onto a 15 cm tissue culture dish (passage 1). Cells from the second passage were plated on a 6-well plate (3 × 10^5^ per well) for stimulation. The medium was refreshed after 24 h with stimulation media for 72 h: (1) control medium—NF medium with 2% of FBS instead of 10%; (2) control medium with 1 ng/mL TGF-β1 (763102; BioLegend, San Diego, CA, USA); (3) conditioned medium from 4T1; (4) conditioned medium from 67NR; and (5) conditioned medium from E0771. Media (1), (2), (3), and (4) were applied on BALB/s NFs, whereas media (1), (2), and (5) were applied on C57BL/6 NFs.

### 2.10. Fluorescence Microscopy Analyses

A total of 2.5 × 10^3^ cells/well were cultured for imaging on a Falcon^®^ 96-well Black/Clear Flat Bottom TC-treated Imaging Microplate (Corning, New York, NY, USA). Staining was carried out after stimulation with TGF-β1 or cancer-cell-conditioned medium. Before staining, cells were washed twice with PBS solution, fixed in freshly prepared 4% (*v*/*v*) paraformaldehyde (Avantor Performance Materials Poland, Gliwice, Poland) for 10–15 min, washed thrice with PBS, and permeabilized in 0.25% (*v*/*v*) Triton X-100 (Sigma-Aldrich, Saint-Louis, MO, USA) for 15 min at room temperature (only wells for VDR staining). After washing thrice in the PBS solution, cells were blocked for 30 min in 1% (*w*/*v*) bovine serum albumin (Sigma-Aldrich, Saint-Louis, MO, USA) solution in 0.1% (*v*/*v*) PBS/Tween 20 (Sigma-Aldrich, Saint-Louis, MO, USA) at room temperature. Then, the fixed cells were rinsed thrice with PBS (5 min each) and incubated with primary antibodies against VDR (bs-2987R, dilution 1:100; Bioss Antibodies, Woburn, MA, USA) and fibroblast activation protein (FAP) (ab28244, dilution 1:500; Abcam, Cambridge, UK) in blocking solution at 4 °C overnight. Next, after washing thrice with PBS, a secondary antibody (Anti-rabbit Antibody Alexa Fluor 488, ab150077; Abcam, Cambridge, UK) in the blocking solution was used for 1 h at room temperature. Then, the samples were rinsed thrice with PBS and stained with DAPI (dilution 1:1500; Cell-Signaling, Danvers, TX, USA) and DyLight™ 554 Phalloidin (dilution 1:100; Cell-Signaling, Danvers, MA, USA) in PBS solution for 15 min at room temperature. Cells were photographed using an Olympus IX81 fluorescence microscope (Olympus, Warsaw, Poland) with CellSense software (Olympus, Warsaw, Poland). Fluorescence was determined from the images using ImageJ according to the protocol of Luke Hammond (QBI, The University of Queensland, Australia; accessed on 18 October 2021; www.theolb.readthedocs.io/en/latest/imaging/measuring-cell-fluorescence-using-imagej.html). The areas of interest (whole cell for FAP staining and cell nucleus for VDR staining) and areas of background were measured, and corrected total cell fluorescence (CTCF) was calculated using the following formula:CTCF = integrated density − (area of the selected cell × mean fluorescence of background readings) 

### 2.11. Real-Time qPCR Analysis of Cultured NFs Isolated from Lungs of Healthy BALB/c or C57BL/6 Mice Fed Control Diet and Vitamin-D_3_-Deficient Diet and/or Treated with Calcitriol

From the stimulated ex vivo NFs, total RNA was extracted using 1 mL Tri-reagent (Sigma Aldrich, Saint-Louis, MO, USA) followed by RNA purification with Direct-zol™ RNA Miniprep (ZYMO RESEARCH, Tustin, CA, USA) according to the manufacturer’s protocol. Using the iScript cDNA Synthesis Kit (Bio-Rad, Hercules, CA, USA), 1 µg of purified RNA was reverse transcribed into complementary DNA (cDNA). Then, the expression of the following genes was determined in cDNA with ready-to-use primers and probes (TaqMan^®^ Gene Expression Assays; Thermo Fisher Scientific, Waltham, MA, USA): *Acta2* (Mm01546133_m1), *Mmp9* (Mm00442991_m1), *Spp1* (Mm00436767_m1), and *Vdr* (Mm00437297_m1). Real-time qPCR was carried out in a 10 µL reaction volume composed of 20× presented probes, 50 ng cDNA, and 2× TaqMan™ Gene Expression Master Mix (Thermo Fisher Scientific, Waltham, MA, USA) in a ViiA™ 7 Real-Time PCR System (Thermo Fisher Scientific, Waltham, MA, USA) using the following program: 10 min at 95 °C for initial denaturation and 40 cycles at 95 °C for 15 s and 60 °C for 1 min. Expression was calculated according to the comparative ΔΔCt method in which *Gadph* (Mm99999915_g1) and *Rps27a* (Mm01180369_g1) were used as endogenous controls and normalized to each untreated control using QuantStudio™ Real-Time PCR Software and ExpressionSuite Software (Thermo Fisher Scientific, Waltham, MA, USA).

### 2.12. Tumor Tissue Preparation for Enzyme-Linked Immunosorbent Assays (ELISA)

Tumor specimens were frozen in liquid nitrogen, suspended in RIPA buffer (Sigma-Aldrich, Saint-Louis, MO, USA) that contained protease and phosphatase inhibitors (both Sigma-Aldrich, Saint-Louis, MO, USA), and then mechanically homogenized (MP Biomedicals, Santa Ana, CA, USA). The samples were centrifuged at 10,000× *g* for 10 min at 4 °C after homogenization, and supernatants were transferred to fresh Eppendorf tubes. Protein concentration in the homogenates was measured using the Quick Start™ Bio-Rad Protein Assay (Bio-Rad, Hercules, CA, USA). Tumor protein samples were used for ELISA.

### 2.13. Quantitative Protein Evaluation by ELISA

ELISA kits were used to assess the expression of selected proteins in mouse plasma and tumor homogenates. Assays were performed according to the manufacturer’s protocols, and absorbance values were measured using a plate reader (BioTek Instruments, Winooski, VT, USA). The expression of the following proteins was measured in plasma: C-C motif chemokine ligand 2 (CCL2), TGF-β (eBioscience, Amsterdam, The Netherlands), IL-6 (Wuhan EIAab Science Co, Ltd., Wuhan, China), OPN (Thermo Fisher Scientific, Waltham, MA, USA), and vascular endothelial growth factor (VEGF; Invitrogen, Carlsbad, CA, USA). The expression of the following proteins was measured in tumor samples: CCL2 (eBioscience, Amsterdam, The Netherlands), fibroblast growth factor 23 (FGF-23; Abcam, Cambridge, UK), IL-6 (Wuhan EIAab Science Co, Ltd., Wuhan, China), OPN (Thermo Fisher Scientific, Waltham, MA, USA), SDF-1, TGF-β, and VEGF (Invitrogen, Carlsbad, CA, USA). The results obtained were analyzed using the CurveExpert ver. 1.4 software.

### 2.14. Statistical Analysis 

Statistical analysis was performed using the GraphPad Prism 7.1 software. Using the Shapiro–Wilk data normality test, the normality of the data distribution was analyzed (assumption of significance of the test for *p* < 0.05). The datasets that did not meet normality requirements (did not pass the Shapiro–Wilk test with α = 0.05) were further tested using the Kruskal–Wallis test using Dunn’s post-test for multiple comparisons (if not otherwise stated in the dataset description). Datasets that met the normality requirement were further analyzed using a one-way ANOVA followed by Sidak’s post-hoc test for multiple comparisons (since only selected, biologically/experimentally meaningful comparisons were challenged Sidak’s post-test provides better correction for multiple comparisons, thus, provides a better test power). The statistical analysis of individual data, depending on their distribution, is presented in table and figure legends. Differences between groups for which *p* < 0.05 were considered statistically significant.

## 3. Results

### 3.1. Impact of Vitamin D on Surface Markers of Lung NFs and CAFs

NFs isolated from the lungs of 4T1 tumor-bearing BALB/c mice (both on normal and on vitamin-D_3_-deficient diet) treated with calcitriol showed significantly increased expression of α-SMA. Compared with the deficiency group, the expression of podoplanin and TNC was increased on fibroblasts from mice fed the vitamin-D_3_-deficient diet and treated with calcitriol. Furthermore, the expression of these two proteins, as well as PDGFRβ, was found to be decreased in mice fed with 5000 IU vitamin-D_3_-supplemented and 100 IU vitamin-D_3_-deficient diets (Table 2). Administration of different types of diets or calcitriol did not change the expression of these proteins on lung fibroblasts from 67NR-bearing BALB/c mice (Table 2). In C57BL/6 mice bearing E0771 tumors, only podoplanin was decreased in mice fed with diets containing 5000 IU vitamin D_3_ and in the 100 IU+cal group (Table 2). Similar characteristics were observed for NFs isolated from healthy BALB/c and C57BL/6 mice, with only a decrease in α-SMA in all treatment groups of BALB/c mice (Appendix A).

Differences in vitamin D status did not change the expression of surface antigens on CAFs derived from 4T1 and 67NR tumors (Table 3). However, an exception was the decreased expression of podoplanin on CAFs isolated from 4T1 tumor-bearing mice fed with 5000 IU of vitamin D_3_ and mice fed with a control 1000 IU diet and administered with calcitriol by gavage (Table 3). A decrease in α-SMA was observed on CAFs from E0771 tumors isolated from the 1000 IU+cal group, similar to that observed for the 5000 IU group. A reverse tendency was observed for podoplanin and PDGFRβ as the expression of these proteins was increased as compared to the control group. Increased expression was also observed for TNC in all groups supplemented with vitamin D or calcitriol (Table 3).

### 3.2. Analysis of Blood Flow in Tumor Tissue from 4T1, 67NR, and E0771 Tumor-Bearing Mice

To evaluate the blood flow in tumor tissue, the following time–intensity curve (TIC) parameters were estimated: peak enhancement (PE)—the maximum intensity in the TIC (blood volume); rise time (RT)—calculated from the beginning of enhancement to PE; mean transit time (mTT)—corresponding to the center of gravity of the best-fit function of echo-power (or fitted signal); wash-in area under the TIC (WiAUC); wash-in rate, maximum slope between the time of onset of contrast inflow and the time of PE on the TIC (WiR); wash-in perfusion index (WiPI = WiAUC/RT)—representative of blood flow, relative blood volume (rBV = amplitude of the plateau + offset amplitude), and relative blood flow (rBF = rBV/mTT) (Table 4, Appendix A).

Both vitamin D_3_ supplementation and calcitriol treatment influenced the perfusion of 4T1 tumors. Significantly reduced WiR was observed in calcitriol-receiving groups. This indicates the substantially reduced blood inflow into tumor tissue, which is confirmed by reduced rBF in those groups. Overall tumor perfusion was also significantly reduced, as indicated by reduced WiAUC and PE. At the same time, supplementation with a high amount of vitamin D_3_ (5000 IU) caused a significant increase in the blood inflow rate (WiR), but a significantly lower WiAUC compared with the 1000 IU group. The high level of WiAUC in the 100 IU group, along with the long RT, indicates the slower blood supply in tumors in the mice fed a vitamin-D_3_-deficient (100 IU) diet, but equally good supply as observed in the 1000 IU reference group. Thus, it can be concluded that in the 4T1 model, vitamin D_3_ supply modulates the kinetic, but not overall, tumor perfusion, whereas calcitriol addition reduces tumor perfusion. This finding is supported by WiPI, which was significantly reduced only in calcitriol-receiving groups (Table 4).

Interestingly, a similar analysis conducted with E0771-bearing mice showed distinct observations. Vitamin D_3_ deficiency (100 IU group) increased the WiR (along with PE and WiAUC), which resulted in an increase in WiPI, indicating high blood perfusion (significantly higher than in the 1000 IU and 5000 IU groups). Surprisingly, calcitriol addition further increased the WiR when combined with 100 IU vitamin D_3_ (with negligible influence when combined with 1000 IU). At the same time, a significant reduction in RT along with a substantial decrease in PE, rBV, and WiAUC, was observed, resulting in the WiPI level comparable with other groups, indicating no changes in tumor perfusion. Thus, it can be concluded that in the E0771 model, vitamin D_3_ deficiency increases tumor perfusion, and calcitriol addition counteracts it (Table 4).

In mice bearing 67NR tumors, no statistically significant changes in blood flow were observed (Appendix A). Altogether, both vitamin D_3_ supply and calcitriol addition affect tumor perfusion differently, depending on cancer cell lines used in the study.

### 3.3. COL1A1 and α-SMA Expression in Tumor Tissue

COL1A1 staining of tumor cells did not change significantly in 4T1 and E0771 tumors, but in 67NR tumors, the deficiency diet led to a significant increase in COL1A1 expression as compared to the control and 100 IU+cal groups (Appendix A). The expression of α-SMA did not change significantly in all tumor models (Appendix A).

### 3.4. Plasma and Tumor Tissue Expression of Selected Cytokines and Growth Factors

The plasma concentrations of OPN were diminished in the 100 IU vitamin-D_3_-deficient group in 4T1-bearing mice and in 100 IU+cal in 67NR tumor-bearing mice, but in healthy BALB/c and C57BL6 mice, the concentration was the highest in the 100 IU+cal group (Table 5). A higher OPN plasma concentration was observed in 4T1 tumor-bearing mice as compared to healthy BALB/c mice and mice bearing 67NR or E0771 tumors. The OPN plasma concentration was also significantly higher in E0771 tumor-bearing mice as compared to C57BL/6 healthy mice (Appendix A). ELISA tests performed in tumor tissue lysates showed that 4T1 tumors were characterized by higher OPN expression as compared to 67NR and E0771, irrespective of the treatment group. No significant changes in OPN expression were observed between the treatment groups with the exception of E0771, where OPN expression in tumor tissue was diminished in the 5000 IU group (Appendix A). The expression of TGF-β was diminished in the plasma of mice bearing 4T1 tumors from 1000 IU+cal, 5000 IU, and 100 IU groups. A similar tendency of TGF-β expression was observed in the plasma of 67NR-bearing mice from the 100 IU group, whereas an opposite tendency was observed in the 1000 IU+cal and 5000 IU groups (Table 5). In C57BL/6 mice bearing E0771 tumors, as well as in healthy mice, the plasma concentration of TGF-β did not change (Table 5). Furthermore, the TGF-β plasma concentration did not differ between tumor models or healthy mice fed with vitamin-D_3_-sufficient diet (1000 IU), whereas in mice bearing E0771 tumors, the expression of TGF-β was significantly higher as compared to 4T1 tumor-bearing mice. Among the 5000 IU groups of BALB/c mice, the concentration of TGF-β was the lowest in 4T1 tumor-bearing mice. However, in deficiency groups (with and without calcitriol), the TGF-β plasma concentration was higher in healthy BALB/c mice as compared to 4T1 tumor- or 67NR tumor-bearing mice (Appendix A). The TGF-β concentration measured in tumor tissue was significantly higher in 67NR tumors from the 100 IU deficiency group as compared to all supplemented groups. A similar tendency was observed in calcitriol-treated 4T1 tumors as compared to the 100 IU-treated group, but the difference was not statistically significant. 4T1 and E0771 tumors showed a similar tumor tissue concentration of TGF-β, but significantly higher than that in 67NR tumors (Appendix A). The plasma concentration of CCL2 in 4T1 tumor-bearing mice was diminished in the 100 IU+cal group as compared to 100 IU and 1000 IU+cal groups, while no change was observed in 67NR tumor-bearing mice and an increased concentration in healthy BALB/c mice in calcitriol-treated groups (Table 5). In C57BL/6 healthy mice, the plasma concentration of CCL2 was diminished in the deficiency group as compared to the control group, and a similar tendency was preserved in E0771 tumor-bearing mice (*p* = 0.0779) (Table 5). The highest plasma concentration of CCL2 was observed in C57BL/6 mice bearing E0771 tumors irrespective of the treatment (as compared to C57BL/6 healthy mice and all BALB/c mice). Among BALB/c mice, the highest CCL2 concentrations were observed in 67NR tumor-bearing (1000 IU and 1000 IU+cal) and healthy (100 IU+cal) mice (Appendix A). An increased plasma concentration of CCL2 was noticed in the tumor tissues of 67NR tumors in the 100 IU group, and in 4T1 tumors from the same group, a tendency to decrease was observed. In E0771 tumors, a significantly decreased CCL2 plasma concentration was observed in the 100 IU+cal group. The CCL2 plasma concentration was higher in E0771 tumors as compared to 4T1 and 67NR (Appendix A). The plasma concentration of VEGF and IL-6 was also analyzed. The VEGF plasma concentration was decreased in 67NR mice in all treatment groups and was almost undetectable with ELISA in BALB/c healthy and C57BL/6 healthy and tumor-bearing mice (Appendix A). The highest tumor tissue concentration of VEGF was observed in E0771-bearing mice (Appendix A). In turn, the tumor tissue concentration of VEGF was not changed significantly in 4T1 or E0771 tumor-bearing mice, while it was found to be increased in 67NR tumors in the 100 IU+cal group as compared to the 1000 IU+cal group. In the 5000 IU group, the VEGF plasma concentration was higher in 4T1 than in 67NR tumors (Appendix A). The IL-6 plasma concentration was changed significantly upon treatments only in healthy BALB/c mice; a significant increase in the IL-6 plasma concentration was observed as compared to the control in the 100 IU+cal group (Appendix A). In mice bearing 4T1 and 67NR tumors, the IL-6 plasma concentration was higher as compared to healthy BALB/c mice in control groups; in the 1000 IU+cal, 100 IU, and 100 IU+cal groups, it was higher only in 4T1 tumors. In the 100 IU+cal group, the IL-6 plasma concentration was higher in E0771 tumor-bearing mice as compared to C57BL/6 healthy mice (Appendix A). SDF-1, the tumor concentration of which was the highest in 4T1 tissue (except for the 100 IU group), did not change significantly with respect to treatments applied (Appendix A). The tumor tissue concentration of FGF23 was significantly increased in 100 IU+cal 4T1 tissue; in 67NR tumors, it was increased in the 5000 IU and 100 IU groups as compared to the control 1000 IU group; and it was diminished in the 100 IU+cal as compared to the 100 IU group. In E0771 tumors, the FGF23 concentration was the lowest in the 100 IU+cal group (Table 5). The FGF23 tumor tissue concentration was higher in E0771 tumors as compared to 4T1 and 67NR and was also higher in 4T1 tissue as compared to 67NR in the 5000 IU and 100 IU+cal groups (Appendix A).

### 3.5. Effect of TGF-β and Cancer Cell CM on Lung Fibroblasts Harvested from Healthy Mice Fed with Different Vitamin D_3_ Diets

Lung fibroblasts obtained from healthy mice fed with different vitamin D_3_ diets and treated or not treated with calcitriol were incubated with TGF-β and CM from 4T1, 67NR, and E0771 cell cultures. Then, the fibroblasts were subjected to further analysis. A comparison of the phenotype analysis of untreated cells is presented in Appendix A.

#### 3.5.1. Flow Cytometry Analyses for α-SMA, Podoplanin, PDGFRβ, and TNC Expression

TGF-β, but not CM from 4T1 or 67NR cells, decreased the expression of α-SMA in fibroblasts harvested from the control group of BALB/c mice. The MFI analysis showed a similar tendency in the remaining groups (Table 6). The effect was more pronounced and statistically significant when considering the percentage of positive staining (Appendix A). Podoplanin expression was downregulated by TGF-β (in all groups of mice) but upregulated by 4T1 CM (in all groups except for the control). On the other hand, it was not influenced by 67NR CM (Table 6). PDGFRβ and TNC were downregulated by both TGF-β and 4T1 CM, but only on fibroblasts obtained from mice fed with the control diet (1000 IU) (Table 6).

TGF-β or E0771 CM did not significantly affect the expression of α-SMA and podoplanin on lung fibroblasts harvested from C57BL/6 mice (Appendix A). In addition, TGF-β or E0771 CM did not influence the expression of TNC (Appendix A), with the exception of fibroblasts from mice fed with the 5000 IU vitamin D diet, where the percentage of TNC-positive cells significantly increased after TGF-β stimulation (Appendix A). In the case of PDGFRβ (Appendix A), especially when considering the percentage of positive stained cells in all groups (Appendix A), TGF-β stimulation led to an increased expression of this receptor. E0771 CM had no significant influence on PDGFRβ expression (Appendix A).

#### 3.5.2. Fluorescence Microscopy Analysis of VDR and FAP Expression in Lung Fibroblasts

The expression of VDR and FAP was also assessed in cultured lung NFs from healthy BALB/c (Figure 2A–L) and C57BL/6 (Figure 3A–J) mice. Lung NFs from BALB/c mice administered with calcitriol showed higher expression of VDR than that of appropriate controls (1000 IU and 100 IU), whereas lung NFs from mice in both vitamin D deficiency and supplementation groups showed lower VDR expression than that of control (1000 IU) mice (Figure 2A). TGF-β stimulation of lung fibroblasts did not change the effect of calcitriol but increased the expression of VDR in fibroblasts from the 5000 IU group (Figure 2B,K). Stimulation with 4T1 CM led to a similar profile of VDR expression as stimulation with TGF-β, but the effect was weaker (Figure 2C,K). Stimulation with 67NR CM did not significantly alter the relationships observed in nonstimulated cells (Figure 2A vs. Figure 2D). However, an analysis of the effects of stimulations used in the control group (1000 IU) revealed that none of them affected VDR expression (Figure 2I). In lung fibroblasts obtained from mice on a control diet and administered with calcitriol (1000 IU+cal), the expression of VDR significantly decreased upon all stimulations used. However, in fibroblasts from the 5000 IU, 100 IU, and 100 IU+cal groups, stimulation with TGF-β and 4T1 CM caused a significant increase in VDR expression. 67NR CM had no effect on VDR expression in these groups (Figure 2I).

FAP expression was the highest in lung fibroblasts from mice on a deficient diet administered with calcitriol (100 IU+cal), and in general, all treatments used resulted in increased expression of FAP in lung fibroblasts derived from BALB/c mice (Figure 2E). Stimulation with TGF-β led to a reduction in or elimination of differences observed in nontreated cells derived from various groups of BALB/c mice. At the same time, TGF-β led to a significant decrease in FAP staining in cells derived from the 1000 IU+cal group as compared to the control (1000 IU) group (Figure 2F). A similar effect was observed in cells incubated with 4T1 CM: a significantly lower expression of FAP in both groups treated with calcitriol and the deficient group (Figure 2G,L). In the case of 67NR CM stimulation, lung fibroblasts from mice administered with calcitriol and vitamin-D-deficient mice showed higher FAP staining than that of control mice. Fibroblasts from mice on the vitamin-D-supplemented diet showed lower FAP staining than control mice upon 67NR CM stimulation (Figure 2H). While comparing the effects of stimulation within each group of mice, it was observed that TGF-β stimulation increased FAP expression in the control (mice fed with the 1000 IU vitamin D diet) and deficiency groups. However, in the deficiency group treated with calcitriol, the opposite effect was observed. Stimulation with 4T1 CM led to increased FAP fluorescence in fibroblasts from control mice and decreased FAP fluorescence in fibroblasts from the 1000 IU+cal, 100 IU, and 100 IU+cal groups. 67NR CM decreased FAP staining in the 5000 IU and 100 IU+cal groups and increased it in the deficient group (Figure 2J,L).

For control mice, the highest VDR expression was observed in cultured ex vivo lung NFs from healthy C57BL/6 mice. Calcitriol administration to the vitamin-D-deficient group led to a significant increase in VDR expression (Figure 3A). Stimulation with TGF-β or E0771 CM did not significantly alter these relationships (Figure 3B,C,I). However, both stimulation with TGF-β and E0771 CM of lung fibroblasts from control mice decreased VDR expression. VDR expression in lung fibroblasts from mice fed with a control diet and treated with calcitriol did not change upon stimulations, but it was increased by TGF-β in the 5000 IU, 100 IU, and 100 IU+cal groups (Figure 3G). FAP expression was the highest in lung fibroblasts from mice fed with the 5000 IU diet and the lowest in the deficiency group administered with calcitriol (100 IU+cal) (Figure 3D,J). Stimulation with TGF-β or E0771 CM exacerbated the effect of calcitriol in lowering the FAP expression in both groups (Figure 3E,F). However, the effect of TGF-β stimulation on FAP expression was opposite to that of E0771 CM stimulation, causing an increase and a decrease in FAP expression, respectively. Therefore, the lowest FAP expression was noticed in fibroblasts harvested from mice fed a deficient diet and administered with calcitriol (100 IU+cal) (Figure 3H,J).

#### 3.5.3. Acta2, Mmp9, Spp1, and Vdr Expression in Cultured NFs Isolated from Lungs of Healthy BALB/c or C57BL/6 Mice Fed with Control and Vitamin-D_3_-Deficient Diet and/or Treated with Calcitriol

The expression of selected genes was measured in the control (1000 IU) and deficiency groups (100 IU and 100 IU+cal), in which the highest effects on fibroblasts were noticed (Figure 4). In NFs from BALB/c mice, the expression of *Acta2* (encoding α-SMA) increased upon stimulation with TGF-β, but this effect was not significant in the 100 IU group (Figure 4A). In C57BL/6 NFs, TGF-β stimulation increased *Acta2* expression in all groups; however, this effect was the lowest in the 100 IU+cal group. E0771 CM also stimulated *Acta2* expression, but only in NFs from mice treated with 100 IU+cal (Figure 4B). In NFs from the BALB/c deficiency group, the expression of *Mmp9* was significantly stimulated by TGF-β and 4T1 CM. In the 100 IU+cal group, this effect was preserved only after stimulation with 4T1 CM (Figure 4A). 4T1 CM also stimulated the expression of *Spp1* (encoding OPN), but only in NFs from control (1000 IU) BALB/c mice (Figure 4A). Interestingly, in NFs from C57BL/6 control mice (1000 IU), decreased *Spp1* expression was observed after stimulation with E0771 CM and TGF-β. In NFs from deficiency group stimulation with E0771 CM led to a significant increase in *Spp1* expression in comparison to the 1000 IU group (Figure 4B). TGF-β stimulation decreased the expression of *Vdr* in NFs from all groups of BALB/c and C57BL/6 mice (Figure 4A,B). In the 1000 IU groups, 4T1 CM increased, while E0771 decreased, the *Vdr* expression. However, in deficiency groups, an increase in *Vdr* expression was observed only after stimulation with 67NR CM. Moreover, *Vdr* expression was the highest after 67NR stimulation in the 100 IU+cal group, similar to that observed after E0771 CM stimulation (Figure 4A,B).

## 4. Discussion

To further understand the effect of varying vitamin D_3_ status on the growth and metastasis of breast cancer, we implanted two metastatic mouse breast cancer cells (4T1 and E0771) and nonmetastatic cells (67NR) into mice fed with vitamin-D_3_-normal, vitamin-D_3_-deficient, and vitamin-D_3_-supplemented diets. Implantation was performed when differences in the plasma 25(OH)D_3_ concentrations statistically differed between normal and experimental diets, which was achieved after 6 weeks of feeding [3]. On day 7 after tumor transplantation, calcitriol gavage (1 µg/kg/day) was started in mice fed with normal and deficient diets and the same diets were continued [3]. As discussed earlier, the effect of the above treatments on tumor tissue blood flow was the highest in deficiency groups in both 4T1 and E0771 tumors: vitamin D_3_ deficiency significantly increased selected blood-flow parameters, whereas a deficient diet followed by calcitriol gavage led to a significant decrease in most of them. However, in 4T1 tumors (not in E0771), some parameters describing the blood flow were also decreased, but other parameters were increased in mice fed with a normal diet and administered with calcitriol. The modulation of tumor vascularization is not only important in the context of metastasis but may also have an impact on the response to treatment. Decreased blood flow in the tumor may cause a decrease in the response to chemotherapy [40]. A previous study showed that calcitriol treatment led to decreased vascularization of MCF-7 human breast cancer tumors with overexpression of VEGF [41]. However, other studies have shown that calcitriol stimulated in vitro expression of VEGF and downregulated the expression of thrombospondin 1. In T47D human breast cancer, calcitriol in vivo inhibited tumor growth, but had no effect on vascular density. Moreover, calcitriol showed a stimulating effect on the plasma VEGF concentration in the tested mice [42]. In the present study, no effect of the used treatment on VEGF expression was observed in 4T1 or E0771 tumor tissue or in mice plasma; however, in 67NR tumor-bearing mice, in which no effect of treatments on blood flow was noted, the VEGF plasma concentration was decreased irrespective of the treatment as compared to control mice. Therefore, it can be concluded that the effect of vitamin D_3_ on tumor vascularization or angiogenic factors may depend on the type of tumor. Our studies have indicated that the basal expression of molecules, which can be implicated in the angiogenesis process, differ significantly between mouse mammary gland tumors and the plasma of tumor-bearing mice. For instance, in this study, the concentration of CCL2 was significantly higher in the tumor-bearing mice and in the plasma of E0771-bearing mice; however, OPN was high in concentration in 4T1 plasma and tumor tissue. Both cytokines can modify the tumor microenvironment, either by directly influencing angiogenesis [43,44] or via recruiting macrophages (CCL2) [45] and activating fibroblasts (OPN) [24]. This effect may also depend on the dose and schedule of treatment with calcitriol or vitamin D_3_ (as can be seen from different effects of calcitriol gavage on, for example, blood flow in tumor tissue in mice fed with vitamin-D_3_-normal or vitamin-D_3_-deficient, or vitamin-D_3_-supplemented diet) as well as the observation time. In our previous studies, a decrease in the 4T1 tumor tissue VEGF concentration was found on day 14 of tumor progression in mice injected subcutaneously (0.5 µg/kg/day) with calcitriol or its analogs, with increased PE and TTP observed in young mice on day 24 [2]. However, the effect on 4T1 cancer metastasis was the same, namely in mice from which tissue samples were collected in the present study [3], as well as in our previous work [2]: in all groups of mice in which calcitriol was administered or vitamin D_3_ content in the diet was increased, an enhanced metastatic process was observed [2,3].

As evidenced from previous studies, CAFs play an important role in invasion and metastasis as well as in angiogenesis of tumors [20,46]. Tumor cells can stimulate stromal fibroblasts to differentiate into CAFs, and the crosstalk between cancer cells and stromal fibroblasts leads to tumor progression [24]. In the present study, the phenotype of CAFs as well as lung NFs from tumor-bearing and healthy mice was analyzed. Upon various vitamin D_3_ or calcitriol treatments, significant changes in the phenotype of CAFs were observed only in E0771 tumors. Interestingly, in lung NFs from healthy C57BL/6 mice, calcitriol did not influence the expression of studied molecules, but in NFs from BALB/c mice, α-SMA expression was decreased in all treatment groups. However, in tumor-bearing mice, the effect of various treatments on NFs was dependent on the tumors implanted. No significant effects were observed on NFs from 67NR nonmetastatic and E0771 metastatic tumors. Whereas in highly metastatic 4T1 tumor-bearing mice, signs of fibroblast activation were observed in both groups of mice administered with calcitriol but were more pronounced in mice on a deficient diet (100 IU+cal). NFs from this group of mice showed increased expression of α-SMA and podoplanin, as well as TNC, which suggests increased activation of NFs [47]. Interestingly, the expression of these proteins was significantly higher in the 100 IU+cal group also as compared to the supplemented group. In this tumor model, metastases were located in the lungs and were significantly increased in both groups treated with calcitriol as well as in the 5000 IU group but by a lower number (count for macrometastases presented earlier [3] as mean number ± standard deviation is as follows—control 40 ± 4; 1000 IU+cal 67 ± 27; 5000 IU 52 ± 11; 100 IU 44.5 ± 15; 100 IU+cal 77 ± 27) [3]. This proves the impact of the mammary gland tumor type on the biological effects of vitamin D_3_. For example, the expression of OPN and SDF-1, which are some of the important molecules secreted from cancer cells stimulating the differentiation of fibroblasts to myofibroblasts [24,48], was the highest in plasma and/or tumor tissue of 4T1 as compared to 67NR and E0771 tumor-bearing mice. In the present study, no significant effects of vitamin D_3_ on plasma OPN were observed, but previously, in later stages of 4T1 tumor progression (days 28 and 33), calcitriol injections led to an increased plasma concentration of OPN [2]. Moreover, increased features of NF activation noted in 4T1 tumor-bearing mice, which were the most visible in the 100 IU+cal group, may also be an effect of the increased tumor tissue concentration of IL-6, a cytokine that may activate fibroblasts through the induction of STAT3 phosphorylation [49]. As a consequence of myofibroblast activation, the increased expression of α-SMA on NFs may result in macrophage recruitment and M2 polarization as well as myeloid-derived suppressor cells (MDSC) and regulatory T lymphocytes (Tregs) recruitment and their differentiation with Th2 polarization [50,51]. Similarly, the expression of podoplanin on fibroblasts may be responsible for M2 polarization and immunosuppression of the tumor microenvironment [52]. In addition, TNC-expressing fibroblasts modulate the recruitment of monocytes/macrophages [53]. Our previous study has shown that treatment of 4T1 tumor-bearing young mice with calcitriol or its analogs (mice on a diet containing a normal concentration of vitamin D_3_) led to an increase in the percentage of monocytes in the blood [54] and an increase in the ratio of inflammatory (Ly6C^high^CXCR1^low^CCR2^+^) spleen monocytes to monocytes with repair and anti-inflammatory properties (Ly6C^low^CXCR1^high^) [27]. Moreover, previous ex vivo studies confirmed that cultured media from 4T1 cells enhanced the M2 polarization stimulated by calcitriol; moreover, M2 macrophages differentiated in the presence of calcitriol stimulated the migration of 4T1 and 67NR cells in vitro [55]. Treg prevalence with Th2 polarization was also observed in calcitriol-treated mice bearing 4T1 tumors [4,54]. Together, these factors lead to the modulation of blood flow and/or increased metastasis as an effect of calcitriol treatment of mice bearing 4T1 tumors observed in our studies [3].

Further analysis of the effects of calcitriol on fibroblast activation was performed on lung NFs from healthy BALB/c and C57Bl/6 mice treated with various diets and/or calcitriol. It is worth emphasizing that NFs from these two strains of mice react differently to various vitamin D_3_ or calcitriol treatments. The differences between these two strains of mice, for example, in the concentration of vitamin D metabolites were already reported. For example, higher plasma concentrations of 25(OH)D_3_, 24,25(OH)_2_D_3_, and 3-epi-25(OH)D_3_ were observed in C57BL/6 than in BALB/c mice in the control (1000 IU), 1000 IU+cal, and 5000 IU groups [3]. In the present study, decreased α-SMA expression on NFs from BALB/c mice was observed in all treatment groups, but no effect was observed in C57BL/6 mice. In both strains of healthy mice, vitamin D_3_ deficiency combined with calcitriol gavage (100 IU+cal) led to an increased plasma concentration of OPN, but CCL2 and IL-6 were increased in these groups only in BALB/c mice; in C57BL/6 mice, the opposite tendency was observed. NFs growing in such microenvironments modified by varying vitamin D_3_ treatments were stimulated ex vivo with TGF-β or cancer cell CM. Because the most significant changes in fibroblasts were observed in deficiency groups, the gene expression in NFs of these mice was also analyzed. However, these ex vivo studies did not fully confirm the observations of lung NF activation in 4T1 tumor-bearing mice. In the ex vivo experimental conditions of the present study, there were limited components of the full microenvironment in which CAFs interact with tumor cells and with TAMs, and their interactions are important for their final phenotype [56]. In NFs from deficiency groups, 4T1 CM stimulated *Mmp9* expression as well. Circulating MMP9 was previously described as inversely correlated with the vitamin D plasma concentration, with the highest concentrations in vitamin-D-deficient healthy adults [57]. Furthermore, E0771 CM stimulated the expression of *Spp1*. Our previous study demonstrated that calcitriol stimulated OPN secretion in vitro in a mouse BALB/3T3 NF cell line [27], and others reported a similar effect of calcitriol on other mouse and human fibroblast cell lines [32]. This effect is dependent on the interaction of VDR with the *Spp1* gene [58]. Moreover, the authors of these studies reported that CAFs and NFs expressed VDR, and the expression of selected genes such as CD82 and S100A4 correlated with stromal VDR expression and clinical outcome in patients with colorectal patients [32]. VDR expression was also analyzed in stimulated lung NFs in our study. In E0771-stimulated NFs, the expression of *Vdr* mRNA was the highest in the 100 IU+cal group, in which the expression of *Spp1* was also the highest, but on the protein level, the expression of VDR was lower in this group of mice as compared to the control. Thus, in this study, VDR expression did not correspond to the observed effects of vitamin D deficiency/supplementation, also when measured in whole tumor tissue, as we previously described [3].

The differences between two metastatic mouse mammary gland cancer models observed in response to vitamin D_3_ can be clearly visualized. Of particular note are the differences in pulmonary NFs responses to different modes of vitamin D_3_ delivery in the 4T1 tumor model. Mice fed with a vitamin-D_3_-supplemented diet (5000 IU) showed decreased expression of proteins responsible for protumoral activity of myofibroblasts (α-SMA, PDGFRβ, and TNC). This effect was accompanied by high plasma concentrations of vitamin D metabolites 25(OH)D_3_ and 24,25(OH)_2_D_3_ at the beginning of tumor progression, but significantly decreased during continued supplementation in the late stages of tumor growth. High concentrations of 3-epi-25(OH)D_3_ were observed during the whole period of observation [3]. In the E0771 tumor model, no change in lung NF activity was observed, but CAFs in mice fed with a vitamin-D-supplemented diet (5000 IU) were activated as podoplanin, TNC, and PDGFRβ increased. In these mice, however, all vitamin D metabolites measured were high during the whole experiment [3]. Thus, it can be noticed that vitamin D_3_ supplementation may lead to decreased or increased fibroblast activation, which is dependent on the tumor model/microenvironment and may also be dependent on the mice strain studied [3]. Interestingly, vitamin D deficiency (which in general does not lead to increased metastatic process, like vitamin D_3_ supplementation or calcitriol administration in the 4T1 model, but shows a tendency for an increased number of micrometastases [3]) also led to decreased NFs activation in the 4T1 model (decreased podoplanin and TNC) and to some signs of increased activation of CAFs in the E0771 model (increased podoplanin). In these deficient mice, a low concentration of vitamin D metabolites was observed during the whole experiment [3]. The most interesting observation is the comparison of calcitriol administration in mice fed with a normal vs. deficient diet, especially in the 4T1 model. Only α-SMA induction on NFs was observed in the former, but in the latter, podoplanin and TNC also increased significantly. Mice fed with a deficient diet and treated with calcitriol showed the lowest concentrations of all vitamin D metabolites, also as compared to mice on a deficient diet [3].

In summary, the impact of vitamin D_3_ supplementation or calcitriol administration on blood flow or CAFs and lung NFs is not visible in less invasive tumor models, such as 67NR, despite induced changes in plasma and/or tumor tissue concentrations of OPN, CCL2, TGF-β, VEGF, and FGF23. Moreover, differences between invasive tumor models were observed in response to the above treatments. In E0771-tumor-bearing mice, the influence on the metastatic process was not visible, but a tendency of tumor growth inhibition was observed in the 5000 IU diet as well as in 100 IU+cal groups [3]. In the latter, a decrease in the blood flow was also observed, which was accompanied by a decrease in FGF23 and CCL2 in tumor tissue, but an increase in TNC on CAFs. Interestingly, similar effects on blood flow (decrease) were also observed in 4T1 tumors from the 100 IU+cal group, but it did not affect the CAF phenotype. At the same time, increasing α-SMA, podoplanin, and TNC on lung NFs and increasing metastasis. 

## 5. Conclusions

Divergent effects of vitamin D_3_ supplementation or deficiency in healthy mice lead to the formation of various microenvironments in the body, which depends on the mouse strain. Tumors develop in such microenvironments and themselves modify the microenvironments, producing higher concentrations of OPN, SDF-1 (4T1), TGF-β (4T1 and E0771), CCL2, VEGF, FGF23 (E0771), and IL-6 (67NR), which influence the response to vitamin D_3_ supplementation/deficiency and calcitriol administration. Based on the results presented here, it can be concluded that the safest approach may be to constantly maintain the appropriate concentration of vitamin D because additional supplementation, especially in the case of primary deficiency, may lead to undesirable effects.

## Figures and Tables

**Figure 1 cancers-14-04585-f001:**
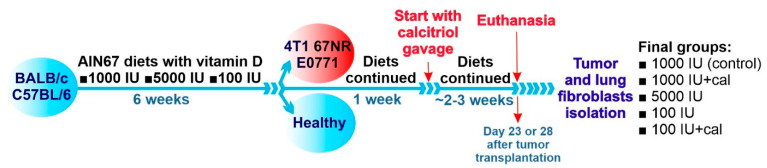
Scheme of the treatment of BALB/c and C57BL/6 mice. Mice of both strains were fed ad libitum with diets containing various amounts of vitamin D_3_ for 6 weeks. Next, on the day assigned as day 0, tumor cells were implanted orthotopically in selected mice (4T1 and 67NR to BABL/c mice, E0771 to C57BL/6 mice). The diet was continued for the next 7 days, and then calcitriol administration per os was started and continued thrice a week till day 28 (for BALB/c mice) or day 23 (for C57BL/6 mice). Calcitriol was administered to mice on control (1000 IU) and deficient (100 IU) diets at a dose of 1 µg/kg. During autopsy, blood, tumors, and lungs were harvested for further analyses.

**Figure 2 cancers-14-04585-f002:**
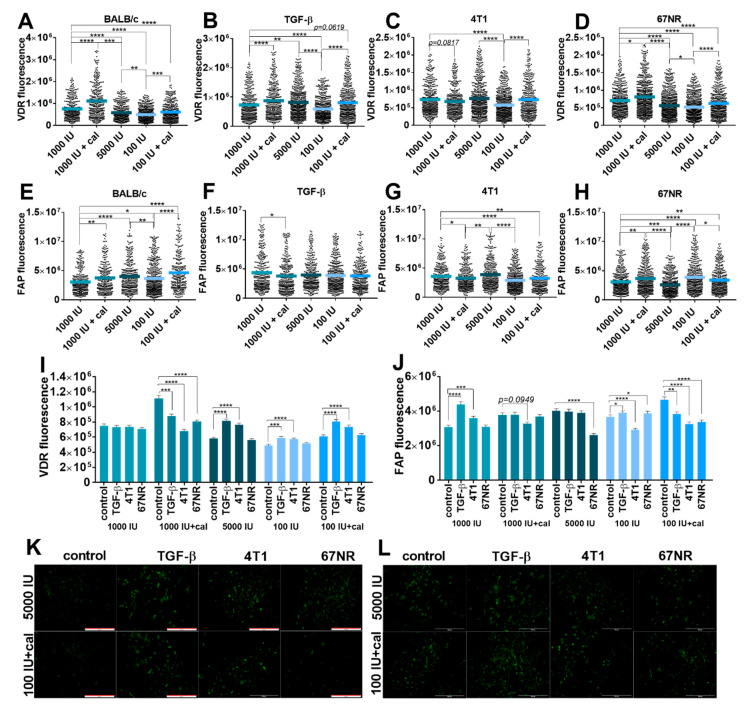
Fluorescence microscopy analysis of vitamin D receptor (VDR) and fibroblast activation protein (FAP) expression in cultured NFs isolated from lungs of healthy BALB/c mice fed various amounts of vitamin D_3_ and/or treated with calcitriol. (**A**–**D**,**I**) VDR expression; (**E**–**H**,**J**) FAP expression. Example images from selected groups are presented for (**K**) VDR and (**L**) FAP. NFs were treated ex vivo with transforming growth factor-β (TGF-β) or culture media from the culture of 4T1 and 67NR cells. Cells were reviewed and photographed using an Olympus IX81 fluorescence microscope. Fluorescence was determined from images using ImageJ according to the protocol of Luke Hammond (QBI, The University of Queensland, Australia; accessed on 18 October 2021; https://theolb.readthedocs.io/en/latest/imaging/measuring-cell-fluorescence-using-imagej.html). The areas of interest (whole cell for FAP staining and cell nucleus for VDR staining) and areas of background were measured, and Corrected Total Cell Fluorescence (CTCF) was calculated using the following formula: CTCF = integrated density–(area of selected cell × mean fluorescence of background readings). Data are shown as (**A**–**H**) mean with individual data; (**I**–**J**) mean ± standard error of mean. Statistical analysis: Kruskal–Wallis test followed by Dunn’s test. * *p* < 0.05; ** *p* < 0.01; *** *p* < 0.001; **** *p* < 0.0001.

**Figure 3 cancers-14-04585-f003:**
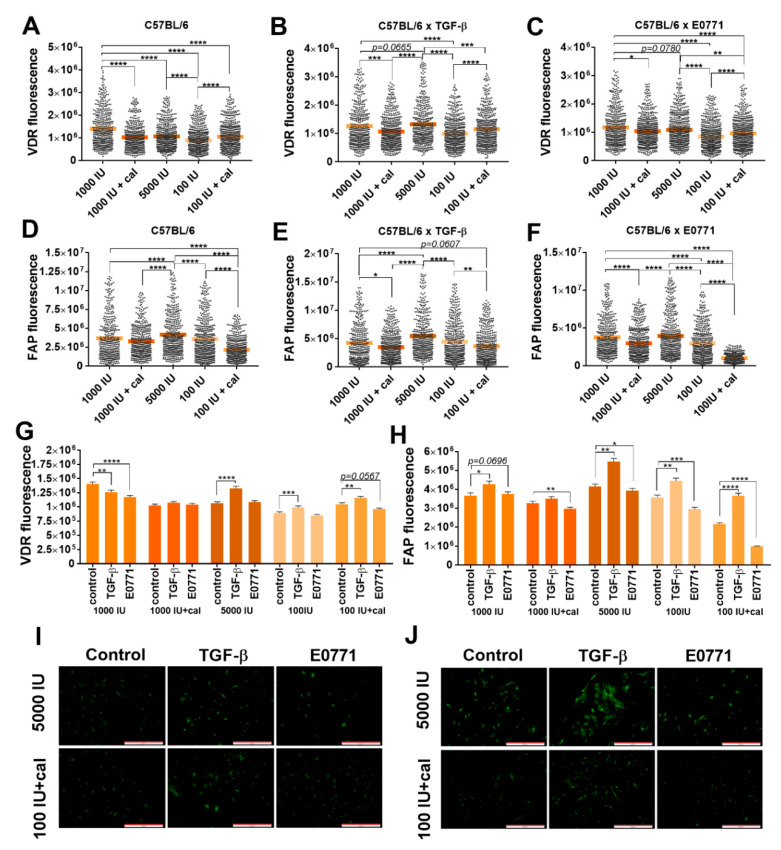
Fluorescence microscopy analysis of vitamin D receptor (VDR) and fibroblast activation protein (FAP) expression in cultured NFs isolated from lungs of healthy C57BL/6 mice fed various amounts of vitamin D_3_ and/or treated with calcitriol. (**A**–**C**,**G**) VDR expression; (**D**–**F**,**H**) FAP expression. Example images from the selected groups were presented for (**I**) VDR and (**J**) FAP. Scale bars = 500 µm. NFs were treated ex vivo with transforming growth factor-β (TGF-β) or culture media from the culture of E0771 cells. Cells were reviewed and photographed using an Olympus IX81 fluorescence microscope. Fluorescence was determined from images using ImageJ according to the protocol of Luke Hammond (QBI, The University of Queensland, Australia; accessed on 18 October 2021; https://theolb.readthedocs.io/en/latest/imaging/measuring-cell-fluorescence-using-imagej.htmL). The areas of interest (whole cell for FAP staining and cell nucleus for VDR staining) and areas of background were measured and Corrected Total Cell Fluorescence (CTCF) was calculated using the following formula: CTCF = integrated density − (area of selected cell × mean fluorescence of background readings). Data are shown as (**A**–**H**) mean with individual data; (**I**–**J**) mean ± standard error of mean. Statistical analysis: Kruskal–Wallis test followed by Dunn’s test. * *p* < 0.05; ** *p* < 0.01; *** *p* < 0.001; **** *p* < 0.0001.

**Figure 4 cancers-14-04585-f004:**
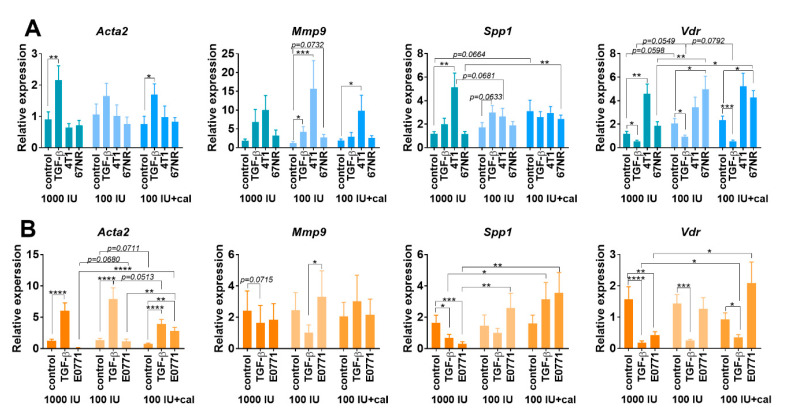
Gene expression in cultured NFs isolated from lungs of healthy BALB/c or C57BL/6 mice fed a control and vitamin-D_3_-deficient diet and/or treated with calcitriol. NFs were treated ex vivo with transforming growth factor-β (TGF-β) or culture media from the culture of 4T1, 67NR (**A**), or E0771 (**B**) cells. Expression of *Acta2* (α-smooth muscle actin), *Mmp9* (matrix metalloproteinase 9), *Spp1* (osteopontin) and *Vdr* (vitamin D receptor) was calculated according to the comparative ΔΔCt method with *Gadph* and *Rps27a* as endogenous controls and normalized to each untreated control using QuantStudio™ Real-Time PCR Software and ExpressionSuite Software. Data are shown as mean ± standard error of mean. Statistical analysis: Kruskal–Wallis test followed by Dunn’s test. * *p* < 0.05; ** *p* < 0.01; *** *p* < 0.001; **** *p* < 0.0001.

**Table 1 cancers-14-04585-t001:** Remmele and Stegner scale: percentage of positive cells (A) and the intensity of the color reaction (B). The final score represents the product of these two values (A × B).

A		B	
Points	Description	Points	Description
0	No cells with a positive reaction	0	No staining
1	Up to 10% cells with a positive reaction	1	Low intensity of staining
2	11–50% cells with a positive reaction	2	Moderate intensity of staining
3	51–80% cells with a positive reaction	3	Intense staining
4	>80% cells with a positive reaction		---

**Table 2 cancers-14-04585-t002:** Flow cytometry analysis of lung NFs from mice bearing 4T1, 67NR, and E0771 tumors.

	Marker	1000 IU	1000 IU+cal	5000 IU	100 IU	100 IU+cal
**4T1**	α-SMA	36.3 ± 36.1	220 ± 56.3 ^ab^	9.9 ± 3.3	23.3 ± 220	218 ± 72.6 ^abc^
	Podoplanin	8.9 ± 5.4	10.8 ± 4.0 ^b^	4.6 ± 1.0	4.6 ± 1,8 ^a^	9.5 ± 3.9 ^bc^
	PDGFRβ	3.9 ± 0.6	3.6 ± 0.7 ^b^	2.6 ± 0.2 ^a^	2.6 ± 0.6 ^a^	3.0 ± 0.6
	TNC	5.0 ± 1.7	4.8 ± 1.1 ^b^	2.7 ± 0.6 ^a^	3.2 ± 1.3	5.1 ± 1.4 ^bc^
**67NR**	α-SMA	14.6 ± 10.3	13.3 ± 6.5	11.2 ± 4.5	13.4 ± 5.3	16.7 ± 12.2
	Podoplanin	3.3 ± 0.7	3.9 ± 0.8	3.8 ± 0.7	3.7 ± 0.9	3.8 ± 1.6
	PDGFRβ	2.6 ± 0.3	2.9 ± 1.0	2.2 ± 1.0	2.6 ± 0.4	2.5 ± 0.4
	TNC	2.6 ± 0.4	2.6 ± 0.5	2.7 ± 0.5	2.7 ± 0.6	2.8 ± 0.6
**E0771**	α-SMA	67.3 ± 23.9	62.9 ± 14.6	53.0 ± 19.9	53.9 ± 14.4	50.9 ± 12.5
	Podoplanin	2.9 ± 0.5	2.9 ± 0.5	2.5 ± 0.4	2.6 ± 0.3	2.5 ± 0.3
	PDGFRβ	1.2 ± 0.1	1.2 ± 0.1	1.1 ± 0.0	1.2 ±0.1	1.1 ± 0.1
	TNC	1.2 ± 0.2	1.2 ± 0.2	1.1 ± 0.1	1.2 ± 0.1	1.3 ± 1.1

α-SMA—α-smooth muscle actin; EpCAM—epithelial cell adhesion molecule; PDGFRβ—platelet-derived growth factor receptor beta; TNC—tenascin C. Data are shown as normalized median fluorescence intensity (MFI) calculated by dividing the MFI of the stained sample by the MFI of isotype control. Steps of CD_3_1^−^EpCAM^−^CD45^−^ fibroblast gating and example dot-plots after staining of CD_3_1^−^EpCAM^−^CD45^−^ fibroblasts with isotype controls and appropriate antibodies are shown in Appendix A. Mice were fed with diets containing various amounts of vitamin D for 6 weeks. Next, on the day assigned as day 0, tumor cells were implanted orthotopically. Diets were continued for the next 7 days, and then calcitriol gavage was started and continued thrice a week till day 23 (for C57BL/6 mice) or day 28 (for BALB/c mice). Calcitriol was administered to mice on control (1000 IU) and deficient (100 IU) diets at a dose of 1 µg/kg. During autopsy, lungs were harvested and NFs were isolated. Flow cytometry analyses were performed on NFs after thawing to determine the expression of α-SMA, podoplanin, PDGFRβ, and TNC. *N* = 5–7. Statistical analysis: Kruskal–Wallis test followed by Dunn’s test for multiple comparisons. Data are shown as mean ± SD. ^a^ ≤ 0.05 as compared to 1000 IU, ^b^ ≤ 0.05 as compared to 5000 IU, ^c^ ≤ 0.05 as compared to 100 IU.

**Table 3 cancers-14-04585-t003:** Flow cytometry analysis of CAFs from mice bearing 4T1, 67NR, and E0771 tumors.

	Marker	1000 IU	1000 IU+cal	5000 IU	100 IU	100 IU+cal
**4T1**	α-SMA	1.4 ± 0.5	0.9 ± 0.2	1.4 ± 1.2	1.7 ± 0.7	1.0 ± 0.4
	Podoplanin	0.5 ± 0.2	0.4 ± 0.1	0.4 ± 0.1	0.4 ± 0.1	0.4 ± 0.1
	PDGFRβ	13.4 ± 10.9	15.7 ± 7.7	15.8 ± 6.0	15.5 ± 6.0	15.1 ±7.5
	TNC	2.5 ± 1.5	1.4 ± 0.6	1.5 ± 1.0	1.7 ± 0.6	1.7 ± 0.7
**67NR**	α-SMA	34.9 ± 10.8	37.5 ± 6.8	27.9 ± 5.8	35.8 ± 4.7	30.6 ± 4.6
	Podoplanin	24.1 ± 6.2	27.9 ± 7.4	19.3 ± 7.6	27.0 ± 10.3	17.7 ± 5.6
	PDGFRβ	0.5 ± 0.1	0.5 ± 0.1	0.5 ± 0.2	0.5 ± 0.1	0.5 ± 0.0
	TNC	1.8 ± 0.3	1.7 ± 0.2	1.8 ± 0.5	1.9 ± 0.4	1.9 ± 0.4
**E0771**	α-SMA	27.6 ± 0.9	17.4 ± 12.3 ^a^	18.8 ± 1.6	21.6 ± 7.6	22.9 ± 3.4
	Podoplanin	0.5 ± 0.3	1.5 ± 1.1	1.6 ± 0.8 ^a^	0.9 ± 0.8	0.8 ± 0.3
	PDGFRβ	0.8 ± 0.1	0.9 ± 0.1	0.9 ± 0.0 ^a^	0.8 ± 0.9	0.8 ± 0.1
	TNC	1.9 ± 0.3	1.5 ± 1.4 ^a^	1.6 ± 0.1 ^a^	1.4 ± 0.2	1.7 ± 0.4 ^a^

α-SMA—α-smooth muscle actin; EpCAM—epithelial cell adhesion molecule; PDGFRβ—platelet-derived growth factor receptor beta; TNC—tenascin C. Data are shown as normalized median fluorescence intensity (MFI) calculated by dividing the MFI of the stained sample by the MFI of the isotype control. Steps of CD_3_1^−^EpCAM^−^CD90^+^ fibroblast gating and example histograms after staining of CD_3_1^−^EpCAM^−^CD90^+^ fibroblasts with isotype controls and appropriate antibodies are presented in Appendix A. Mice were fed with diets containing various amounts of vitamin D for 6 weeks. Next, on the day assigned as day 0, tumor cells were implanted orthotopically. Diets were continued for the next 7 days, and then calcitriol gavage was started and continued thrice a week till day 23 (for C57BL/6 mice) or day 28 (for BALB/c mice). Calcitriol was administered to mice on control (1000 IU) and deficient (100 IU) diets at a dose of 1 µg/kg. During autopsy, tumors were harvested and CAFs were isolated. Flow cytometry analyses were performed to determine the expression of α-SMA, podoplanin, PDGFRβ, and TNC. *N* = 4. Statistical analysis: Kruskal–Wallis test followed by Dunn’s test for multiple comparisons. Data are shown as normalized MFI ± SD. ^a^ ≤ 0.05 as compared to 1000 IU.

**Table 4 cancers-14-04585-t004:** Blood flow in tumor tissue from 4T1 and E0771 tumor-bearing mice.

	Parameter Measured	1000 IU	1000 IU+cal	5000 IU	100 IU	100 IU+cal
**4T1**	PE [a.u.]	29.8 ± 11.9	20.0 ± 17.5	47.0 ± 36.3 ^b^	35.2 ± 18.9	14.1 ± 6.3 ^c^
	RT [s]	26.5 ± 15.5	15.7 ± 79	19.5 ± 14.0	34.5 ± 13.8	13.1 ± 9.5 ^d^
	WiR [a.u.]	9.6 ± 5.7	1.6 ± 1.6 ^a^	13.9 ± 11 ^b^	3.3 ± 2.2 ^c^	2.2 ± 1.5 ^c^
	WiAUC [a.u.]	602 ± 279	100 ± 71.1 ^a^	225 ± 98.3 ^a^	815 ± 48.7 ^c^	161 ± 83.2 ^ad^
	WiPI [a.u.]	22.5 ± 8.8	8.7 ± 6.7 ^a^	28.0 ± 13.8	24.6 ± 16.2	10.6 ± 5.0 ^cd^
	rBV [a.u.]	6.7 ± 3.2	19.7 ± 9.6 ^a^	16.3 ± 7.9	21.3 ± 16.8 ^a^	7.8 ± 6.0 ^d^
	rBF [a.u.]	1.9 ± 0.5	1.3 ± 1.4	2.6 ± 1.9	1.0 ± 0.8 ^c^	0.6 ± 0.4 ^c^
	MTT [s]	4.7 ± 1.4	45.7 ± 43.4 ^a^	5.7 ± 5.0	11.4 ± 8.9	14.8 ± 14.6
**E0771**	PE [a.u.]	29.2 ± 13.0	30.0 ± 12.3	38.2 ± 24.4	64.2 ± 35.4 ^a^	24.9 ± 16.1 ^d^
	RT [s]	18.9 ± 9.1	12.3 ± 3.3	10.2 ± 0.9 ^a^	11.0 ± 5.0	3.8 ± 2.0 ^ab^
	WiR [a.u.]	4.8 ± 2.2	5.4 ± 1.9	5.9 ± 1.2	9.1 ± 5.0	13.2 ± 7.5 ^abc^
	WiAUC [a.u.]	317 ± 295	314 ± 199	300 ± 238	868 ± 469 ^ac^	158 ± 140 ^d^
	WiPI [a.u.]	23.3 ± 12.0	23.4 ± 10.7	28.3 ± 20.0	55.1 ± 17.7 ^ac^	18.1 ± 11.1 ^d^
	rBV [a.u.]	8.9 ± 5.6	7.9 ± 1.9	12.3 ± 5.1	17.4 ± 4.2 ^a^	3.4 ± 3.8 ^d^
	rBF [a.u.]	1.6 ± 1.2	1.8 ± 0.7	0.7 ± 1.2	1.7 ± 1.2	0.8 ± 0.7
	MTT [s]	4.2 ± 0.8	4.8 ± 2.1	29.4 ± 19.1 ^ab^	15.1 ± 13.0	3.9 ± 2.3 ^c^

Data shown as peak enhancement (PE) representing the maximum intensity in the TIC (blood volume), rise time (RT) calculated from the beginning of enhancement to PE, mean transit time (mTT) corresponding to the center of gravity of the best-fit function of echo-power (or fitted signal), wash-in area under the TIC curve (WiAUC), wash-in rate, maximum slope between the time of onset of contrast inflow and the time of PE on the TIC (WiR), wash-in perfusion index (WiPI = WiAUC/RT)—representing blood flow, relative blood volume (rBV = amplitude of the plateau + offset amplitude), and relative blood flow (rBF = rBV/mTT). Representative pictures of all parameters are presented in Appendix A. Mice were fed with diets containing various amounts of vitamin D_3_ for 6 weeks. Next, on the day assigned as day 0, tumor cells were implanted orthotopically. Diets were continued for the next 7 days, and then calcitriol gavage was started and continued thrice a week. Calcitriol was administered by gavage to mice on control (1000 IU) and deficient (100 IU) diets at a dose of 1 µg/kg. Blood-flow parameters were measured on day 21 (4T1) or on day 19 (E0771). *N* = 4–7. Statistical analysis: analysis of variance test followed by Sidak’s test for multiple comparisons. Data are shown as mean ± SD. ^a^ ≤ 0.05 as compared to 1000 IU, ^b^ ≤ 0.05 as compared to 1000 IU+cal, ^c^ ≤ 0.05 as compared to 5000 IU, ^d^ ≤ 0.05 as compared to 100 IU.

**Table 5 cancers-14-04585-t005:** Expression of selected cytokines in plasma and tumor tissue of BALB/c and C57BL/6 healthy and tumor-bearing mice.

		Cytokine	1000 IU	1000 IU+cal	5000 IU	100 IU	100 IU+cal
**BALB/c**	**4T1**	OPN	608 ± 360	875 ± 602	688 ± 366	478 ± 256 ^b^	676 ± 427
		TGF-β	167 ± 120	14.7 ± 12.8 ^a^	8.2 ± 8.9 ^a^	52.2 ± 113 ^a^	101 ± 179
		CCL2	137 ± 36.8	166 ± 46.2	153 ± 86.6	165 ± 40.1	107 ± 14.1 ^bd^
		FGF23	101 ± 38.5	125 ± 34.7	130 ± 28.0	133 ± 18.6	157 ± 30.9 ^a^
	**67NR**	OPN	79.0 ± 36.8	79.8 ± 12.3	101 ± 31.2	74.0 ± 12.9	58.2 ± 15.7 ^c^
		TGF-β	118 ± 179	269 ± 214	259 ± 190 ^a^	16.3 ± 18.9 ^c^	32.0 ± 61.2 ^c^
		CCL2	238 ± 103	282 ± 135	202 ± 85.5	251 ± 142	293 ± 87.5
		FGF23	37.3 ± 13.9	41.7 ± 13.0	52.3 ± 10.8 ^a^	57.0 ± 5.3 ^ab^	41.9 ± 9.4 ^d^
	**Healthy mice**	OPN	18.0 ± 4.6	15.4 ± 1.1	19.4 ± 10.2	21.0 ± 4.5	26.8 ± 6.9 ^ab^
		TGF-β	138 ± 255	188 ± 222	212 ± 216	2253 ± 3307	1259 ± 2072
		CCL2	95.8 ± 20.2	171 ± 38.6 ^a^	101 ± 23.0 ^b^	130 ± 50.0	252 ± 151 ^acd^
**C57Bl/6**	**E0771**	OPN	83.2 ± 61.0	74.0 ± 39.1	45.8 ± 11.7	50. 6 ± 19.2	76.9 ± 37.6
		TGF-β	190 ± 292	1310 ± 2948	799 ± 1784	1353 ± 2011	1671 ± 2119
		CCL2	1513 ± 1021	1207 ± 870	941 ± 356	734 ± 515	680 ± 222
		FGF23	535 ± 90.5	649 ± 210	503 ± 70.7	495 ± 197	351 ± 35.7 ^abd^
	**Healthy mice**	OPN	26.6 ± 9.4	23.1 ± 3.7	28.7 ± 5.8	25.5 ± 6.5	33.6 ±12.9 ^b^
		TGF-β	70.4 ± 0.0	1188 ± 2515	1001 ± 2278	1142 ± 2584	1031 ± 1855
		CCL2	512 ± 152	446 ± 186	353 ± 109	266 ± 94.7 ^a^	291 ± 91.1

Plasma concentration of osteopontin (OPN), transforming growth factor β (TGF-β), and C-C motif chemokine ligand 2 (CCL2) in BALB/c and C57BL/6 mice. Tumor tissue concentration of fibroblast growth factor 23 (FGF23). Mice were fed with diets containing various amounts of vitamin D_3_ for 6 weeks. Next, on the day assigned as day 0, tumor cells were implanted orthotopically. Diets were continued for the next 7 days, and then calcitriol gavage was started and continued thrice a week till day 23 (for C57BL/6 mice) or day 28 (for BALB/c mice). Calcitriol was administered by gavage to mice on control (1000 IU) and deficient (100 IU) diets at a dose of 1 µg/kg. During autopsy, blood and tumors were harvested. *N* = 5–7. Statistical analysis: Kruskal–Wallis test followed by Dunn’s test for multiple comparisons. Data are shown as mean ± SD. ^a^ ≤ 0.05 as compared to 1000 IU, ^b^ ≤ 0.05 as compared to 1000 IU+cal, ^c^ ≤ 0.05 as compared to 5000 IU, ^d^ ≤ 0.05 as compared to 100 IU.

**Table 6 cancers-14-04585-t006:** Phenotypic characteristics of NFs from healthy BALB/c mice treated ex vivo with TGF-β or CM from the culture of 4T1 and 67NR cells.

Marker	Group	Treatment			
		Control	TGFβ	4T1 CM	67NR CM
α-SMA	1000 IU	74.7 ± 8.9	52.6 ± 14.8 ^a^	66.8 ± 10.5	70.0 ± 15.5
	1000 IU+cal	60.4 ±12.6	46.9 ± 11.2	62.7 ± 9.6	68.0 ± 19.2
	5000IU	57.1 ± 8.5 ^x^	46.6 ± 14.5	67.1 ± 7.9	60.6 ± 13.1
	100 IU	56.9 ± 14.0 ^x^	47.8 ± 9.0	59.5 ± 7.6	66.6 ± 13.5
	100 IU+cal	55.5 ± 15.4 ^x^	45.9 ± 12.5	57.0 ± 16.9	55.8 ± 15.9
Podoplanin	1000 IU	583 ± 135	399 ± 76.9 ^a^	662 ± 119	595 ± 177
	1000 IU+cal	564 ± 152	370 ± 29.4 ^a^	696 ± 145	540 ± 114
	5000IU	535 ± 95.0	349 ± 113 ^a^	670 ± 123 ^a^	620 ± 121
	100 IU	564 ± 77.8	356 ± 97.0 ^a^	708 ± 135 ^a^	601 ± 114
	100 IU+cal	584 ± 125	333 ± 49.5 ^a^	741 ± 171	615 ± 171
PDGFRβ	1000 IU	4.5 ± 0.4	3.6 ± 0.4 ^a^	3.7 ± 0.9 ^a^	4.3 ± 0.5
	1000 IU+cal	4.2 ± 1.1	3.4 ± 0.5	3.6 ± 0.8	4. 2 ± 0.8
	5000IU	3.9 ± 0.7	3.5 ± 0.6	3.7 ± 0.7	4.0 ± 0.7
	100 IU	3.8 ± 0.9	3.1 ± 0.4	3.4 ± 0.7	3.8 ± 0.8
	100 IU+cal	3.3 ± 1.0	3.1 ± 0.5	3.4 ± 1.3	4.2 ± 1.3
TNC	1000 IU	3.4 ± 0.4	2.8 ± 0.3 ^a^	2.9 ± 0.5 ^a^	3.1 ± 0.4
	1000 IU+cal	3.1 ± 0.6	2.7 ± 0.4	3.0 ±0.5	3.0 ± 0.5
	5000IU	3.0 ± 0.6	2.7 ± 0.5	3.1 ± 0.5	3.0 ± 0.5
	100 IU	3.0 ± 0.5	2.5 ± 0.3	2.9 ± 0.5	2.9 ± 0.5
	100 IU+cal	3.0 ± 0.6	2.5 ± 0.4	2.9 ± 0.7	3.2 ± 0.8

α-SMA—α-smooth muscle actin; PDGFRβ—platelet-derived growth factor receptor beta; TNC—tenascin C. Data are shown as normalized median fluorescence intensity (MFI) calculated by dividing the MFI of the stained sample by the MFI of the isotype control. Healthy mice were fed with diets containing various amounts of vitamin D for 7 weeks. Then, calcitriol gavage was started and continued thrice a week till day 28. Calcitriol was administered to mice receiving control (1000 IU) or deficient (100 IU) diets at a dose of 1 µg/kg. During autopsy, lungs were harvested and NFs were isolated. Flow cytometry analyses were performed after the culture of NFs with TGF-β, 4T1, and 67NR culture media to determine the expression of smooth muscle α-actin (α-SMA), podoplanin, platelet-derived growth factor receptor β (PDGFRβ), and tenascin C (TNC). Representative dot-plots are shown in Appendix A. *N* = 6–7. Statistical analysis: one-way analysis of variance test followed by Sidak’s test for multiple comparisons. Data are shown as mean ± SD ^a^ ≤ 0.05 as compared to untreated control in the same mice group (1000 IU, 1000 IU+cal, 5000 IU, 100 IU, or 100 IU+cal), ^x^ ≤ 0.05 as compared to the same treatment in the 1000 IU group.

## Data Availability

All data supporting the results are contained in the manuscript and Appendix A.

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
