# Peer review of "Modulation of Fibroblast Activity via Vitamin D3 Is Dependent on Tumor Type—Studies on Mouse Mammary Gland Cancer"

_cancers, 2022, doi:10.3390/cancers14194585_

Round 1
Reviewer 1 Report
In the study, the authors have made an attempt to study the effects of calcitriol and vitamin D3 deprivation or supplementation on tumor and lung fibroblasts in three different mouse mammary gland cancer models.
The authors have isolated the fibroblast from the mice and have done EIisa assays of different fibrotic factors.
The limitation of the paper is that they have not used other commonly used techniques to prove their hypothesis.
Hence, it would be more convincing to show the outcomes of the studies using western blot assay and immunostaining (IHC / IFC).
Please show the immunostaining for aSMA in the CAFs.
Reviewer 2 Report
This study was conducted to investigate vitamin D supplementation in healthy mice and mice bearing three mouse mammary gland cancers. Overall the paper was well written. Methods are adequately described. However, I have a few concerns and recommendations.
Major
1) I did not see anywhere that the study was approved by the Animal Subjects Ethics Committee known as Institutional Animal Care and Use Committee (IACUC). This approval must happen prior to data collection.
2) Abstract needed a few more details. It is too brief. At least it should have the dose of vitamin D supplementation and a proper conclusion. I realize that there is a word limit. You can eliminate the first sentence safely and is not necessary.
3) In the introduction at least some human studies should have been mentioned. I understand it is an animal study. It is important to connect the findings to humans after all we do animal studies so that the findings can be extrapolated or applied to humans. There are several human studies that found some beneficial effects of vitamin D in cancer. Here are some references:
doi: 10.1016/j.clnesp.2018.12.085
doi: 10.3390/jpm12060944. PMID: 35743729
doi: 10.1016/j.hemonc.2020.08.005 Epub 2020 Sep 26. PMID: 33002425
4) Statistical analysis: Please describe what statistical tests were used on what variables. They stated that the normality test was applied. But they did not mention whether the data are normal or not normal (what variables are normal and what variables are not normal). Just stating that it is described in the figure legends is not enough. All statistical tests used on all variables must be described here in this section in addition to describing in the figure legends or table footnotes.
5) Statistics: Why Sidak’s test was for multiple comparisons? Please justify this in the statistical analysis section. Why not Bonferroni or Tukeys tests? Overall, this section is very not well described.
6) Results: What was the intake of dietary vitamin D and their body weight gain during the study or at the end of the study?
7) What is the justification for why those 3 doses were chosen?
8) Too many figures. The figures are too small and are very difficult to read. For example, if I want to know the "Normalized MFI" for 4T1 cell lines (Figure 1), I have to guess it but I would not know the exact data value. This is the problem with bar diagrams.
So, I recommend that the authors convert these A, B, and C figures into one grand table where the data are accurately reported. Instead of giving **, they can use superscripts a, b, c, etc., for multiple comparisons.
9) Convert Figure 3, A, B, and C, Figure 4, A and B, Figure 5 (a to G), and Figure, 6 (A-D) into four tables as described above. Also, it is very difficult to read the standard deviation (I am assuming it is SD). The rest of the figures are fine.
10) Tables are stand-alone and self-sustaining. That means a reader should understand the data presented in a table without referring to the text of the manuscript. So, please add more footnotes at the bottom of the table with appropriate superscripts embedded in the text of the table. Also, the table footnote should contain a type of statistical test used, abbreviations used, whether the data were mean ± SD or SE, and the significance level.
) 11) Like Tables, figures are also stand-alone. Please give a brief title (legend) at the bottom of the figure. Also, describe the data presented in the figure in a few sentences next to the title of the figure as one paragraph. This should also include data presented (mean±SD or mean±SE), sample size, statistical tests used, or significance level.
10) All abbreviations should be spelled out in the first mention in the abstract, in the text of the manuscript, and in the table/figure. Beginning from the second occurrence, the abbreviation should be used.
11) Discussion: Please remove the results from the discussion. No need to repeat results in the discussion.
Minor:
1) change "sacrificed" to "killed".
2) There are too many acronyms used. Maybe those are necessary. If a phrase or word is used 4 times, no need to use an abbreviation, just use full expansion. Only use an abbreviation if a phrase/word appears more than 4 times
Reviewer 3 Report
Modulation of fibroblast activity via vitamin D3 is dependent on tumor type—studies on mouse mammary gland cancer
Cancers-1842349
In this paper, Łabędź et al. analyzed the effects of calcitriol treatment and vitamin D3 (cholecalciferol) deprivation or supplementation on tumor and lung fibroblasts in three mouse mammary gland cancer models (4T1, 67NR, and E0771) with different metastatic potential. Generally speaking, most of the claims were supported by the data. However, to me it is just piling up data rather conveying a key finding, which I hope the authors can improve for the revision.
1. What is the most important finding from your work? Right now for me it is all over the place. Some data are interesting, however, all lacks deep dive to delineate the molecular mechanism.
2. Please focus one or two key discoveries and dig deeper to provide a mechanistical understanding. This is really an open-ended comment however in my view is critical to allow acceptance of the paper.
3. There are typos throughout the manuscript, wrong labeling of sub-headings, and numerous inconsistency in fonts of text.
Round 2
Reviewer 1 Report
The authors response is not satisfactory.
Author Response
The authors response is not satisfactory.
We assume that the dissatisfaction with our response to reviewer 1's previous comments was because we did not include additional studies on IHC-assessed alpha-SMA expression. Therefore, in the current version of the manuscript, we have included this data in Figure S6 E-H. The immunohistochemical reactions for Alpha-SMA antibody were performed according to the protocol described in this publication. The Smooth Muscle Actin (ready-to-use, 20 min. incubation, cat. no. IR611, Dako, Glostrup, Denmark) antibody was used. As in the case of cytometric analysis (Table 3), also here we did not observe significant changes in the expression of this protein in tumors.
Reviewer 2 Report
I appreciate the changes authors made. The quality has improved significantly. However, there are a few mandatory changes to be made before it goes to the next step.
1) Remove the first sentence from the simple summary.
"Although vitamin D3 and its analogs are known to modulate the activity of fibroblasts under various disease conditions, their impact on cancer associated fibroblasts (CAFs) is yet to be fully investigated."
2) Some sentences are too long and it requires English editing by a native English speaker.
The paper is too wordy. There are so many unnecessary words and phrases everywhere.
3) Authors wrote this:
".......healthy mice fed vitamin D3-normal (1000 IU), deficient (100 IU), and supplemented (5000 IU) diets."
1000/100/5000 IU makes no sense. Are these per day or per kg of diet? Please specify this throughout the paper wherever there were mentioned.
4) Change 'levels' to 'concentrations'. Level is not scientifically appropriate term and should not be used.
5) I liked the introduction linking to human studies.
However the following statement is incorrect. Please correct it.
"Moreover, high 25(OH)D plasma levels (above 100 nmol/L) association with a greater risk of breast cancer in postmenopausal women was also reported [14, 15]."
Please see the conclusions from these 2 references:
Reference #14: In conclusion, serum vitamin D concentrations ≥100 nmol/L are associated with reduced risk of breast cancer in postmenopausal women.
Reference #15: "We confirmed previous findings suggesting that a low 25(OH)D status is associated with an inferior breast cancer survival, but unlike previous findings, we found an indication of poorer breast cancer survival also among women with high 25(OH)D levels."
6) Formatting issues with the table on page 7 to 11. I have no idea what this table is trying to convey. Authors needed to review the PDF/Word before they approve the file for submission.
Is Table 2 necessary? I think this can be explained in the text format in a couple of sentences. As such there are so many graphics/tables in the paper.
7) "euthanized" is not appropriate term. Euthanization is a mercy killing, which is not the case here.
Just simply say, animals were killed. That is exactly what happened. I don't understand the problem of writing "killed" here.
8) I don't understand why so much description was given on the supplementary material. "Supplementary material" should be put in the Supplementary section not in the main manuscript. So please create a link (or journal will create link for you, not sure) and place the link on the front page or in the declarations section (where you list conflict of interest).
As such the paper is so lengthy and you lose the readers.
9) Remove the last figure (Summary, Figure 5). Results have been already described in the results section so detail. You have summarized the findings in the last paragraph, and again you have presented the summary in the form of a figure. This is a overkill. This is triple redundancy.
By doing the same thing multiple times leads to reader's fatigue.
10) Thanks for transferring data from the figure to the Table. It is so much easier to read the data from the table than form the figure.
On the other note, Figure 2 data were transferred into the Table 3 and to the Supplementary material. Then why the Figure 2 is still there in the manuscript. This is redundant.
11) Figure 4 to Table 5: Please see my comment above #10.
12) Figure 5 to Table 6: Please see my comment above #10.
13) Figure 6 to Table 7: Please see my comment above #10.
14) Table footnotes: Please mention what the data are? Please say this in the footnote:
Data are presented as mean ± SD. I am not sure whether you have used SD or SE. I am guessing these are SDs. In case if you used SE, please convert to SD because when you have a small sample size, it is better to report SDs.
15) Tables have too many numbers. Please simplify the data.
For the data with 1 to 2 digit numbers, use 1 decimal. For 3 or more digit numbers, no need to use decimals at all. When the data are less than 1, then one can use up to 2 decimals. Please use this as an example to fix all the data that were presented in the text and in tables.
16) When you resubmit the paper, please do not use "Track changes". Just highlight the revisions in color font. Track changes interferes with reviewing and takes so much time to review
Reviewer 3 Report
It has been improved compared to the previous version. I think it is ok to accept for publication.
Author Response
Thank you
Round 3
Reviewer 2 Report
The authors have put in good work in revising the paper. I just have a few minor recommendations.
Why vitamin D level was not changed to concentration/s? "Level" is not appropriate to use. Please use "concentrations". Any biomarker, if you quantify, it is "concentration" not level.
Combine all 3 paragraphs in the statistical analysis section into one paragraph.
In general there are too many short paragraphs (one sentence) in the paper. Please consolidate these small paragraphs with the paragraph before or after.
Author Response
Thank you very much for your meaningful comments on our manuscript. The changes made as a result improved the clarity of the manuscript. Below are the answers to each question:
Point.1. Why vitamin D level was not changed to concentration/s? "Level" is not appropriate to use. Please use "concentrations". Any biomarker, if you quantify, it is "concentration" not level.
Response 1. Thank you for this comment. Throughout the manuscript, we changed the terminology used from level to concentration or expression, where this was more appropriate.
Point 2. Combine all 3 paragraphs in the statistical analysis section into one paragraph.
In general there are too many short paragraphs (one sentence) in the paper. Please consolidate these small paragraphs with the paragraph before or after.
Response 2. We have merged all the excessively fragmented paragraphs into larger ones, which now seems to increase the clarity of the manuscript.
We have put all the above-mentioned changes into the manuscript by highlighting them in yellow. Currently, however, it is not possible to upload the manuscript to the editorial page until the other reviewers have given their opinion.